# Context-aware Learned Mesh-based Simulation via Trajectory-Level Meta-Learning

**Philipp Dahlinger**  *philipp.dahlinger@kit.edu*
*Autonomous Learning Robots, Karlsruhe Institute of Technology, Karlsruhe*

**Niklas Freymuth**  *niklas.freymuth@kit.edu*
*Autonomous Learning Robots, Karlsruhe Institute of Technology, Karlsruhe*

**Tai Hoang**  *tai.hoang@kit.edu*
*Autonomous Learning Robots, Karlsruhe Institute of Technology, Karlsruhe*

**Tobias Würth**  *tobias.wuerth@kit.edu*
*Institute of Vehicle System Technology, Karlsruhe Institute of Technology, Karlsruhe*

**Michael Volpp**  *michael.volpp@de.bosch.com*
*Autonomous Learning Robots, Karlsruhe Institute of Technology, Karlsruhe*
*Bosch Center for Artificial Intelligence*

**Luise Kärger**  *luise.kaerger@kit.edu*
*Institute of Vehicle System Technology, Karlsruhe Institute of Technology, Karlsruhe*

**Gerhard Neumann**  *gerhard.neumann@kit.edu*
*Autonomous Learning Robots, Karlsruhe Institute of Technology, Karlsruhe*

**Reviewed on OpenReview:** *https://openreview.net/forum?id=j5uACS2Doh*

## Abstract

Simulating object deformations is a critical challenge across many scientific domains, including robotics, manufacturing, and structural mechanics. Learned Graph Network Simulators (GNSs) offer a promising alternative to traditional mesh-based physics simulators. Their speed and inherent differentiability make them particularly well suited for applications that require fast and accurate simulations, such as robotic manipulation or manufacturing optimization. However, existing learned simulators typically rely on single-step observations, which limits their ability to exploit temporal context. Without this information, these models fail to infer, e.g., material properties. Further, they rely on auto-regressive rollouts, which quickly accumulate error for long trajectories. We instead frame mesh-based simulation as a trajectory-level meta-learning problem. Using Conditional Neural Processes, our method enables rapid adaptation to new simulation scenarios from limited initial data while capturing their latent simulation properties. We utilize movement primitives to directly predict fast, stable and accurate simulations from a single model call. The resulting approach, Movement-primitive Meta-MESHGRAPHNET (M3GN), provides higher simulation accuracy at a fraction of the runtime cost compared to state-of-the-art GNSs across several tasks.

## 1 Introduction

The simulation of complex physical systems is crucial to a wide variety of engineering disciplines, including solid mechanics (Yazid et al., 2009; Zienkiewicz & Taylor, 2005; Stanova et al., 2015), fluid dynamics (Chung, 1978; Zienkiewicz et al., 2013; Connor & Brebbia, 2013), and electromagnetism (Jin, 2015; Polycarpou, 2022; Reddy, 1994). In particular, the simulation of object deformations under external forces finds widespread

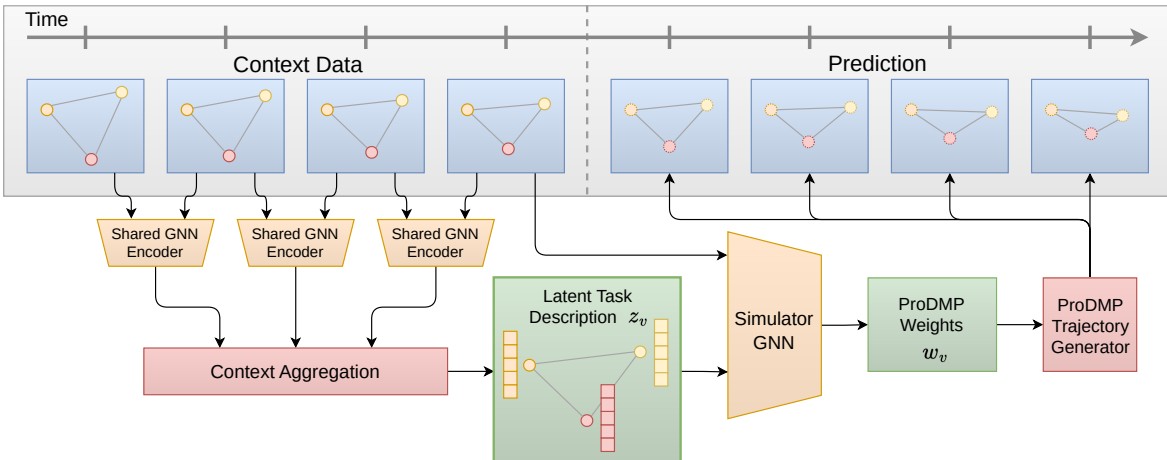

Figure 1: Movement-Primitive Meta-MeshGraphNet (M3GN) aims to learn accurate simulation dynamics from a few initial observations, enabling it to infer material properties from limited historical data. Given a context set of initial system states, node-level latent features are computed for every pair of states using a shared Graph Neural Network (GNN) encoder. These features are then aggregated to form a node-level latent task description $z_v$. This description is concatenated with the last system state to predict Probabilistic Dynamic Movement Primitive (ProDMP) weights, which are used to compute per-node trajectories.

application in, e.g., robotic applications (Scheikl et al., 2022; Wang & Zhu, 2023; Linkerhägner et al., 2023). Mesh-based simulations are appealing for such problems due to the computational efficiency and accuracy of the underlying finite element method (Brenner & Scott, 2008; Reddy, 2019). However, the diversity of the problems to be modeled usually demands task-specific simulators in order to accurately capture the relevant physical quantities (Reddy & Gartling, 2010). Such specialized simulators can be slow and cumbersome to use, especially for large-scale simulations (Paszynski, 2016; Hughes et al., 2005).

Thus, data-driven surrogate models trained on reference simulations have become an appealing alternative (Guo et al., 2016; Da Wang et al., 2021; Li et al., 2022). Among them, general-purpose Graph Network Simulators (GNSs) have recently become increasingly popular (Battaglia et al., 2018; Pfaff et al., 2021; Allen et al., 2022b; 2023; Lippe et al., 2023; Linkerhägner et al., 2023; Yu et al., 2024; Würth et al., 2025). GNSs encode the simulated system as a graph of interacting entities whose dynamics are predicted using GNNs (Bronstein et al., 2021). GNS are one to two orders of magnitude faster than classical simulators (Pfaff et al., 2021) while being fully differentiable. These properties make them highly effective for, e.g., inverse design problems (Allen et al., 2022b; Xu et al., 2021), robotics (Shi et al., 2023; Hoang et al., 2025), and other engineering applications that benefit from fast simulation surrogates (Simon, 2021). However, existing GNS-based models rarely exploit historical context, limiting their ability to infer material properties or long-term system behavior.

Consider, for example, a robotic manipulation task, where the robot needs to maintain an accurate model of some deformable object with unknown material properties to achieve a certain goal (Antonova et al., 2022; Shi et al., 2023; Hoang et al., 2025). Current learned models rely mainly on the current observation, preventing them from using the historical context to infer knowledge about the material or other relevant behaviors (Linkerhägner et al., 2023). To alleviate this issue, we propose to provide a few initial mesh states to learned simulators, allowing the model to observe how the given system behaves before having to simulate subsequent steps. Equipped with this knowledge, the model may infer latent material properties, facilitating for more accurate simulations that precisely extend the previous context. To the best of our knowledge, we are the first to study this specific experimental setup.

This setting naturally fits into a meta-learning framework (Schmidhuber, 1992; Thrun & Pratt, 1998; Vilalta & Drissi, 2005; Hospedales et al., 2022), where the sequence of initial states induces the unique task of predicting its subsequent simulated trajectory. Concretely, we employ Conditional Neural Processs (CNPs) (Garnelo et al., 2018a) to aggregate the provided context sets and the dynamics inferred from them into a latent

descriptor, which is then used to predict the rest of the trajectory. This formulation reveals opportunities to better adapt the standard GNS training setup to meta-learning scenarios, particularly when models must condition on initial context.

GNSs are typically trained through simple next-step supervision (Battaglia et al., 2018; Pfaff et al., 2021; Allen et al., 2023). During inference, entire trajectories are simulated by iteratively predicting per-node dynamics from an initial system state in an autoregressive manner, only ever considering the previous system state or a fixed-size history. This approach is prone to error accumulation over time, decreasing accuracy for longer time horizons (Brandstetter et al., 2022; Han et al., 2022; Radler et al., 2025).

Since the initial context set influences the full trajectory, we decide to utilize trajectory-level predictions, where our model consumes the context steps and directly outputs the remaining simulation in a single forward pass. This formulation offers several advantages over alternatives such as recurrent architectures (Hochreiter & Schmidhuber, 1997; Ruiz et al., 2020; Mienye et al., 2024) or multi-step training (Shi et al., 2023). By predicting the entire trajectory in a single forward pass, we improve training stability and memory efficiency by avoiding gradient updates across multiple passes. This single-pass approach also increases inference speed and completely sidesteps the error accumulation common in autoregressive rollouts.

To this end, we employ node-level Probabilistic Dynamic Movement Primitives (ProDMPs) (Schaal, 2006; Paraschos et al., 2013; Li et al., 2023), which allow us to output complete trajectories using only the latent descriptor as inputs. By representing trajectories with basis functions, ProDMPs neither require autoregressive rollouts, which is costly and error-prone for long horizons, nor large memory for constructing temporal convolutions, unlike prior works (Dahlinger et al., 2025; Xu et al., 2024; Cini et al., 2025). These components enable efficient training and rapid adaptation to trajectory-specific simulation parameters, such as unknown material properties, without requiring explicit parameter knowledge during training or inference.

The resulting method, called Movement-Primitive Meta-MESHGRAPHNET (M3GN), allows the generation of context-dependent simulation trajectories that accurately infer and integrate unknown system properties. Figure 1 provides an overview of our approach, while Figure 2 shows examples for different tasks. To validate the effectiveness of M3GN, we adapt existing experiment suites (Linkerhägner et al., 2023; Dahlinger et al., 2025) to our trajectory-based meta-learning setup, and additionally introduce two novel tasks based on challenging deformable object simulations with varying object materials. Our results show that our method provides superior simulation accuracy compared to several variants of MESHGRAPHNET (MGN) (Pfaff et al., 2021) and recent trajectory-based learned simulators (Xu et al., 2024; Dahlinger et al., 2025)[1]. Further, M3GN's ProDMPs trajectory representation reduces the number model calls at inference, improving inference runtime by up to 32 times compared to MGN.

In summary, we propose M3GN, a novel GNS that (i) extracts a latent descriptor from initial states and uses it to generate the remaining node-level trajectory in a single shot, while rapidly adapting to varying material properties; (ii) achieves state-of-the-art inference speed by coupling CNP-based context aggregation with a physically consistent trajectory formulation via ProDMPs; and (iii) surpasses recent GNSs on challenging deformation benchmarks, yielding superior long-horizon accuracy and stability. To support our claims, we introduce a new experimental setup in which the initial part of each trajectory serves as context data for the model.

## 2 Related Work

**Graph Network Simulators.** Deep neural networks for physical simulations can provide significant speedups over traditional simulators while being fully differentiable (Pfaff et al., 2021; Allen et al., 2022a), making them a natural choice for applications like model-based Reinforcement Learning (Mora et al., 2021) and Inverse Design problems (Baqué et al., 2018; Durasov et al., 2021; Allen et al., 2022a). A popular class of learned neural simulators are Graph Network Simulators (GNSs) (Battaglia et al., 2016; Sanchez-Gonzalez et al., 2020). GNSs utilize Message Passing Networks (MPNs), a special type of GNN (Scarselli et al., 2009; Bronstein et al., 2021) that representationally encompasses the function class of many classical solvers (Brandstetter et al.,

---

[1]Code is provided in the supplement. Here, the reviewers can also find videos showing the qualitative results of M3GN predictions and comparisons to existing baselines.

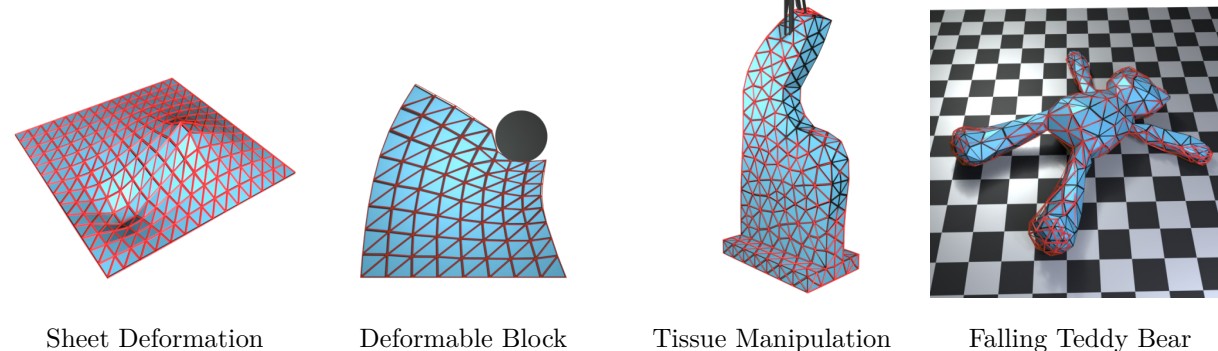

Sheet Deformation      Deformable Block      Tissue Manipulation      Falling Teddy Bear

Figure 2: Final M3GN simulation steps for different evaluation tasks. From left to right: a sheet deforms under two orthogonal forces, a falling collider deforming a block in 2D, a surgical tool dragging tissue, and a falling teddy bear. All visualizations present the **predicted mesh** (blue) alongside a reference **wireframe** (red) of the ground-truth simulation. M3GN takes the deformable object positions from a few initial time steps to predict the remaining simulation steps using per-node movement primitives.

2022). GNS handle physical data by modeling arbitrary entities and their relations as a graph. Applications of GNSs include particle-based simulations (Li et al., 2019; Sanchez-Gonzalez et al., 2020; Whitney et al., 2023), atomic force prediction (Hu et al., 2021), and fluid dynamic problems (Brandstetter et al., 2022). These models have additionally been applied to the mesh-based prediction of deformable objects (Pfaff et al., 2021; Weng et al., 2021; Han et al., 2022; Fortunato et al., 2022; Linkerhägner et al., 2023). Recent extensions handle rigid objects (Allen et al., 2022b; 2023; Lopez-Guevara et al., 2024) and integrate learned adaptive meshing strategies (Plewa et al., 2005; Freymuth et al., 2023; 2025) into the simulator (Wu et al., 2023).

Existing work that considers unknown material properties in simulations of deformable objects combines the GNS prediction with point cloud information to improve long-term predictions Linkerhägner et al. (2023). This method requires a constant stream of point clouds to ground the simulation in, but can not aggregate this information into a description of the material properties. Additionally, the DEL method (Wang et al., 2024) integrates physical priors from the Discrete Element Analysis (DEA) framework with learnable graph kernels, addressing the challenges of simulating 3D particle dynamics from 2D images.

In the context of larger-scale simulations, foundation models are gaining traction in neural simulation tasks, as exemplified by Aurora (Bodnar et al., 2024), a large-scale model trained on extensive climate data. While Aurora demonstrates impressive performance on atmospheric predictions, including global air pollution and weather forecasts, it requires significantly more data for fine-tuning compared to our approach, which focuses on efficient adaptation with fewer data points. Notably, all previously mentioned GNSs predict system dynamics iteratively from a given state, whereas we directly estimate entire trajectories, improving rollout stability and reducing function calls. Related to our approach is the Equivariant Graph Neural Operator (EGNO) (Xu et al., 2024), which also predicts full trajectories using SE(3) equivariance to model 3D dynamics and capture spatial and temporal correlations.

All previously discussed approaches rely on supervised learning or fine-tuning large foundation models, whereas we employ meta-learning to enable efficient adaptation to new simulation conditions. Recently, the Meta Neural Graph Operator (MaNGO) (Dahlinger et al., 2025) introduced meta-learning into Graph Network Simulators. Our work differs in several key aspects: MaNGO requires access to simulation parameters at least for the training set, while our method remains fully parameter-agnostic during both training and inference. Moreover, the meta-learning setup in MaNGO uses different simulation trials with the same material properties as context, whereas we exploit the temporal evolution within a single simulation to infer latent material characteristics. Finally, MaNGO's trajectory representation is less memory-efficient, as it does not employ a compact formulation such as ProDMPs.

**Meta-Learning.** Meta-learning (Schmidhuber, 1992; Thrun & Pratt, 1998; Vilalta & Drissi, 2005; Hospedales et al., 2022) extracts inductive biases from a training set of related tasks in order to increase data efficiency

on unseen tasks drawn from the same task distribution. In contrast to other multi-task learning methods, such as transfer learning (Krizhevsky et al., 2012; Golovin et al., 2017; Zhuang et al., 2020), which merely fine-tune or combine standard single-task models, meta-learning makes the multi-task setting explicit in the model architecture (Bengio et al., 1991; Ravi & Larochelle, 2017; Andrychowicz et al., 2016; Volpp et al., 2019; Santoro et al., 2016; Snell et al., 2017). This explicit architecture allows the resulting meta-models to learn *how* to learn new tasks from a small number of example contexts. A popular variant is Model-Agnostic Meta-Learning (MAML) (Finn et al., 2017; Grant et al., 2018; Finn et al., 2018; Kim et al., 2018), which employs standard single-task models and formulates a multi-task optimization procedure.

Neural Processes (NPs) (Garnelo et al., 2018a;b; Kim et al., 2019; Gordon et al., 2019; Louizos et al., 2019; Volpp et al., 2021; 2023) instead build on a multi-task model architecture (Heskes, 2000; Bakker & Heskes, 2003) but employ standard gradient based optimization algorithms (Kingma & Ba, 2015; Kingma & Welling, 2014; Rezende et al., 2014; Zaheer et al., 2017). Here, we use Conditional Neural Processes (CNPs) (Garnelo et al., 2018a), which aggregate learned features over a variable-sized context set to yield a latent task description that our downstream GNS is conditioned on. Compared to regular NPs, CNPs assume a deterministic task description, eliminating the need for a distribution over latent variables. This assumption simplifies and accelerates the training process, as our objective is to predict a single precise simulation trajectory from the context set. While epistemic uncertainty cannot be fully eliminated, the simulation data itself is deterministic and lacks aleatoric noise, making a distributional latent formulation less appealing and inconsistent with the probabilistic assumptions that motivate Neural Processes.

## 3 Movement-Primitive Meta-MeshGraphNets

In this section, we present the theoretical foundation of the M3GN method, detailing the algorithmic design choices that guided its development.

**Graph Network Simulators.** Consider a graph $\mathcal{G} = (\mathcal{V}, \mathcal{E}, \mathbf{X}_{\mathcal{V}}, \mathbf{X}_{\mathcal{E}})$ with nodes $\mathcal{V}$, edges $\mathcal{E}$, and associated vector-valued node and edge features $\mathbf{X}_{\mathcal{V}}$ and $\mathbf{X}_{\mathcal{E}}$. An MPN (Sanchez-Gonzalez et al., 2020; Pfaff et al., 2021) consists of $M$ message passing steps, which iteratively update the node and edge features based on the graph topology. Each such step is given as

$$\mathbf{h}_e^{m+1} = f_{\mathcal{E}}^m(\mathbf{h}_v^m, \mathbf{h}_e^m),$$
$$\mathbf{h}_v^{m+1} = f_{\mathcal{V}}^m(\mathbf{h}_v^m, \bigoplus_{e \in \mathcal{E}_v} \mathbf{h}_e^{m+1}),$$

where $\mathbf{h}_v^m$ and $\mathbf{h}_e^m$ denote embeddings of the system state per node and edge at message passing iteration $m$, respectively. $\mathcal{E}_v \subset \mathcal{E}$ are the edges connected to $v$. Further, $\bigoplus$ denotes a permutation-invariant aggregation operation such as the sum, the max, or the mean. The functions $f_{\mathcal{V}}^m$ and $f_{\mathcal{E}}^m$ are learned Multilayer Perceptrons (MLPs). The network's final output are the node-wise learned representations $\mathbf{h}_v := \mathbf{h}_v^M$ that encode local information of the initial node and edge features.

Conventional GNSs encode the state of the simulated system as a graph, feed it through the MPN, and interpret the per-node outputs as velocities or accelerations (Pfaff et al., 2021). These dynamics are used to forward the simulation in time using, e.g., a forward-Euler integrator (Sanchez-Gonzalez et al., 2020). The graph encodes relative distances and velocities between entities instead of absolute ones, as the resulting equivariance to translation improves generalization (Sanchez-Gonzalez et al., 2020). GNSs usually minimize a next-step Mean Squared Error (MSE) per node during training, adding carefully tuned implicit denoising strategies (Pfaff et al., 2021; Brandstetter et al., 2022) to stabilize long-term predictions. During inference, they compute trajectories by iteratively predicting and integrating their output in an autoregressive fashion. If some simulated objects, like the collider, are known, only the remaining nodes are predicted. Our method instead uses a ProDMP to predict a compact representation of a whole trajectory per system node.

**Probabilistic Dynamic Movement Primitives.** Movement Primitives (MPs) (Schaal, 2006; Paraschos et al., 2013) allow for compact and smooth trajectory representations $y$ via a set of basis functions parameterized by a set of weights $\boldsymbol{w}$. This temporal smoothness is highly beneficial for, e.g., robotic applications (Li et al., 2024; Otto et al., 2022). Recent methods integrate MPs with neural networks to enhance their

expressive capabilities (Seker et al., 2019; Bahl et al., 2020; Li et al., 2023). Dynamic Movement Primitives (DMPs) (Schaal, 2006) use a spring-damper dynamical system governed by parameters $\alpha$ and $\beta$. To manipulate the trajectory, an external forcing term $f$ is added, before the system converges to a predefined goal $g$:

$$\tau^2 \ddot{y} = \alpha \left( \beta(g - y) - \tau \dot{y} \right) + f(x), \quad f(x) = x \boldsymbol{\varphi}^\mathsf{T} \boldsymbol{w}. \tag{1}$$

Here, $\tau$ influences execution speed, while $f$ depends on the basis functions $\boldsymbol{\varphi}$ in force-space, the weights $\boldsymbol{w}$ and the exponential decaying phase $x$. Solving this equation typically is computationally intensive, particularly when the gradient $dy/d\boldsymbol{w}$ is required (Bahl et al., 2020). ProDMPs (Li et al., 2023) instead solve Equation (7) with pre-computed basis functions $\boldsymbol{\Phi}$ in position-space as

$$y(t) = c_1 y_1(t) + c_2 y_2(t) + \boldsymbol{\Phi}(t)^\top \boldsymbol{w}.$$

The term $c_1 y_1(t) + c_2 y_2(t)$ only depends on the initial conditions $[y(t_0), \dot{y}(t_0)]$. ProDMPs thus generate smooth trajectories at an arbitrary temporal resolution from low-dimensional weights $\boldsymbol{w}$. They crucially allow for efficient gradient computation, and can respect different initial conditions such as positions or velocities. We provide an extensive mathematical background of ProDMPs in Appendix A.

**Meta-Learning and Graph Network Simulators.** To enable generalization across tasks with varying properties, we frame GNS as a meta-learning problem. In this setup, each task corresponds to a simulation of a deformable object with unknown material properties. The goal is to learn a simulator that can adapt quickly to a specific scenario using a limited amount of context data. Following the notation of Volpp et al. (2021), the meta-dataset $\mathcal{D} = \mathcal{D}_{1:L}$ consists of simulation trajectories $\mathcal{D}_l = \{\mathcal{G}_{l,1} \ldots \mathcal{G}_{l,T}\}$, where $T$ is the trajectory length. Each simulation step $\mathcal{G}_{l,t} = (\boldsymbol{m}_{l,t}, \boldsymbol{u}_{l,t})$ represents a graph capturing both the deformable object mesh $\boldsymbol{m}_{l,t}$ (describing its position and topology) and an optional rigid collider $\boldsymbol{u}_{l,t}$. Physical proximity is used to define graph edges that model interactions between the deformable object and the collider. At test time, the first $T^c \ll T$ simulation frames, $\mathcal{G}_{l,1:T^c}$, are observed as a context set to predict the remaining trajectory. Following prior work (Pfaff et al., 2021), we assume access to the full collider trajectory during prediction[2], resulting in the complete *context set*:

$$\mathcal{D}_l^c = \{\mathcal{G}_{l,1}, \ldots \mathcal{G}_{l,T^c}\} \cup \{\boldsymbol{u}_{l,T^c+1}, \ldots, \boldsymbol{u}_{l,T}\}. \tag{2}$$

To provide a clear reference point for discussion, we define the *anchor time step* as the final time step $T^c$ of the context set. The corresponding *anchor graph*, $\mathcal{G}_{l,T^c}$, represents the system's state at this point and serves as the starting state for trajectory prediction by the GNSs. Notably, the anchor graph $\mathcal{G}_{l,T^c}$ alone does not capture the complete system state, as the material properties of the deformable object remain unknown. These properties must be inferred from the prior simulation steps, $\mathcal{G}_{l,1:T^c}$, to enable accurate simulation predictions. Figure 3 illustrates this setup.

**Model Architecture.** Our model architecture, M3GN, is designed to learn from context data and predict future simulation steps by leveraging a combination of graph network simulation and meta-learning techniques. The architecture consists of two parts: the computation of the latent task description from the context data and the actual graph network simulation of future simulation steps. We base our context processing on the Conditional Neural Process (CNP) (Garnelo et al., 2018a), as it efficiently encodes a latent description over tasks given a set of context observations. Omitting the task index $l$ to avoid clutter, CNPs expect a context set $\{(\boldsymbol{x}_1, \boldsymbol{y}_1), \ldots, (\boldsymbol{x}_{T^c}, \boldsymbol{y}_{T^c})\}$ consisting of inputs $\boldsymbol{x}_t$ and corresponding targets $\boldsymbol{y}_t$. We translate our context set $\mathcal{D}^c$ from Equation 2 to this format by using each graph $\mathcal{G}_t$ as an input, and setting its labels as the node-wise velocities. This approach allows the model to focus on dynamics rather than absolute positions, which are more task-specific. Assuming a forward-Euler integration scheme with a time step of 1, we numerically approximate the velocities as the difference between the node positions of two consecutive simulation steps. The input graph $\boldsymbol{x}_t$ represents the simulation state at time step $t$, including mesh and collider, while $\boldsymbol{y}_t$ encodes the change in positions between consecutive time steps.

$$\boldsymbol{x}_t = \mathcal{G}_t, \qquad \boldsymbol{y}_t = \text{pos}(\mathcal{G}_{t+1}) - \text{pos}(\mathcal{G}_t).$$

---

[2]This assumption is commonly satisfied in practical scenarios. For instance, in robotic planning tasks, the robot generates multiple candidate future end-effector trajectories and requires predictions of the resulting deformation to select the most suitable plan.

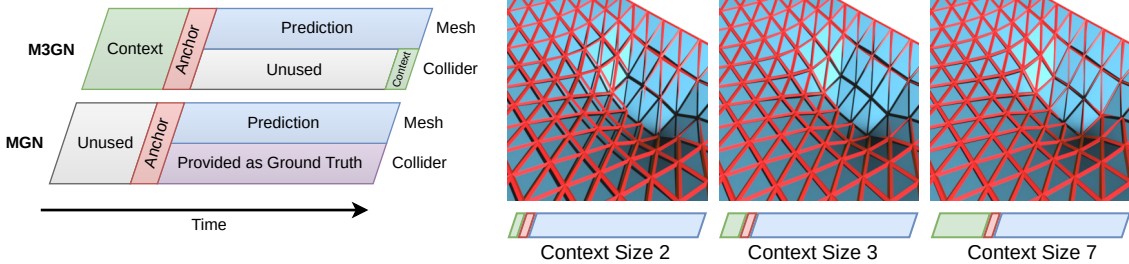

Figure 3: **Left:** M3GN and MGN task setup. Both methods predict mesh positions based on the initial mesh at the anchor time step. M3GN utilizes previous mesh positions and the last step of the collider trajectory for its latent task description, whereas MGN disregards past information and integrates the ground truth collider trajectory into its step-based model. **Right:** Exemplary final simulation steps on the Sheet Deformation task of M3GN given different context set sizes. A larger context size results in a more accurate **prediction** (blue) of the **ground truth** (red wireframe) simulation.

To account for the known collider trajectory, we add its relative position as an additional node feature. Specifically, we include the position of the collider at the last time step $T$ of the simulation, $\mathrm{pos}(\boldsymbol{u}_T)$, relative to its current position, $\mathrm{pos}(\boldsymbol{u}_t)$. Preliminary testing indicated that incorporating the complete future collider trajectory $\mathrm{pos}(\boldsymbol{u}_{T^c}),...\mathrm{pos}(\boldsymbol{u}_T)$ did not improve the results on our tasks. Given a context set $\mathcal{D}^c$ with anchor time step $T^c$, this results in $T^c - 1$ tuples $(\boldsymbol{x}_t, \boldsymbol{y}_t)$. A shared GNN encoder $h_{\boldsymbol{\theta}}$ computes node-level latent features

$$\boldsymbol{z}_{t,v} = h_{\boldsymbol{\theta}}(\boldsymbol{x}_t, \boldsymbol{y}_t) \in \mathbb{R}^{T^c \times |\mathcal{V}| \times d_z} \tag{3}$$

for each context time step with feature dimension $d_z$. We then aggregate over the time domain of the context set to obtain $\boldsymbol{z}_v = \bigoplus_t \boldsymbol{z}_{t,v} \in \mathbb{R}^{|\mathcal{V}| \times d_z}$, using $\bigoplus = \max$ as the aggregation operator. Intuitively, $\boldsymbol{z}_v$ is a representation of the task inferred from the context data $\mathcal{D}^c$, and encodes material properties, future collider movements, and high-level deformations of the simulation. While we use node-level latent features for M3GN, one could additionally aggregate over the nodes to obtain a graph-global task descriptor $\boldsymbol{z} = \bigotimes \boldsymbol{z}_v \in \mathbb{R}^{d_z}$. We explore this choice and different aggregation functions $\bigoplus, \bigotimes$ in Section 4.

Once the task descriptor has been computed, it serves as the input to the predictive stage, enabling simulation of future trajectories. We concatenate the latent description $\boldsymbol{z}_v$ with the node features of the anchor graph $\mathcal{G}_{T^c}$ and subsequently use a GNN $g_{\boldsymbol{\theta}}$ to predict per-node ProDMP weights

$$\boldsymbol{w}_v = g_{\boldsymbol{\theta}}(\mathcal{G}_{T^c}, \boldsymbol{z}_v) \in \mathbb{R}^{|\mathcal{V}| \times d_w}. \tag{4}$$

GNNs provide the flexibility to incorporate arbitrary node features. To better capture short-term dynamics, we augment the input with node velocities from the anchor time step $(T^c)$; the inclusion of velocity information was determined through hyperparameter optimization on a per-task basis. The ProDMP trajectory generator $f(\boldsymbol{w}_v) \in \mathbb{R}^{T \times |\mathcal{V}| \times d_{\mathrm{world}}}$ transforms the predicted outputs of the simulator GNN into per-node object trajectories over the entire simulation horizon. This approach can be seen as a form of temporal bundling (Brandstetter et al., 2022), requiring a single function call. In comparison, existing GNS train mostly on next-step dynamics and require one call per step during their auto-regressive inference scheme (Pfaff et al., 2021; Allen et al., 2023). The trajectory-level view further allows us to omit noise injection during training, which MGN requires to generalize from learned next-step predictions to multi-step rollouts during inference. We provide a visualization of our model architecture in Figure 1 and refer to Appendix B for further details.

**Meta Training.** The goal of meta-learning is to automatically encode inductive biases towards the task distribution extracted from the meta-dataset $\mathcal{D}$ into the task-global parameter $\boldsymbol{\theta}$. To this end, we minimize the negative conditional log probability (Garnelo et al., 2018a)

$$\mathcal{L}(\boldsymbol{\theta}) = -\mathbb{E}_{l \sim 1:L}\left[\mathbb{E}_{T^c \sim T_{\min}:T_{\max}}\left[\log p_{\boldsymbol{\theta}}(\mathrm{pos}(\boldsymbol{m}_{l,1:T}) \mid \mathcal{D}_l^c)\right]\right]. \tag{5}$$

Each training batch consists of a task $\mathcal{D}_l$ for which we sample a context size $T^c$ uniformly between $T_{\min}$ and $T_{\max}$ to ensure that the model learns to handle different context set sizes. We then compute the latent task

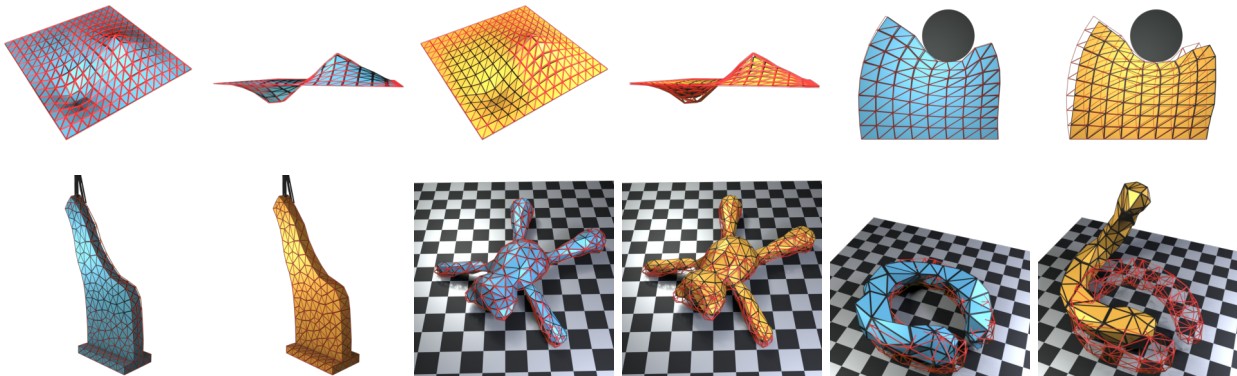

Figure 4: Comparison of the final simulation step between **M3GN** (blue) and **MGN** (orange) on all datasets. From (**Left**) to (**Right**): (**Top**) *Sheet Deformation* and *Deformable Block* with an anchor time step of 2. (**Bottom**) *Tissue Manipulation* with a context size of 5, and *Falling Teddy Bear* and *Mixed Objects Falling* with an anchor time step of 20. M3GN provides much better alignment to the **ground truth** (red wireframe) simulation on all tasks, except for *Tissue Manipulation*, where MGN also solves the task well.

descriptor $\boldsymbol{z}_v$ and subsequently the predicted node trajectories $f(g_{\boldsymbol{\theta}}(\mathcal{G}_{l,T^c}, \boldsymbol{z}_v))$ as described in Equation 3 and Equation 4. The likelihood $p_{\boldsymbol{\theta}}$ is defined to be the Gaussian

$$p_{\boldsymbol{\theta}}(\text{pos}(\boldsymbol{m}_{l,1:T}) \mid \mathcal{D}_l^c) = \mathcal{N}(\text{pos}(\boldsymbol{m}_{l,1:T}) \mid f(g_{\boldsymbol{\theta}}(\mathcal{G}_{l,T^c}, \boldsymbol{z}_v)), \boldsymbol{\sigma}_o). \tag{6}$$

Since the training simulations are not affected by noise, we are not modeling the output variance and set it to $\boldsymbol{\sigma}_0 = 1$. Together with taking the mean over the nodes and time steps to stabilize training, optimizing the Gaussian log likelihood from Equation 6 is equivalent to minimizing the MSE

$$\log p_{\boldsymbol{\theta}}(\text{pos}(\boldsymbol{m}_{l,1:T}) \mid \mathcal{D}_l^c) \simeq \frac{1}{T \, |\mathcal{V}| \, d_{\text{world}}} \sum_{t,v,i} \left( \text{pos}(\boldsymbol{m}_{l,t})_{v,i} - f(g_{\boldsymbol{\theta}}(\mathcal{G}_{l,T^c}, \boldsymbol{z}_v))_{t,v,i} \right)^2.$$

The whole architecture is trained end-to-end using the loss $\mathcal{L}(\boldsymbol{\theta})$ from Equation 5. After the meta-training, we fix $\boldsymbol{\theta}$, which now encodes inductive biases towards the meta-data $\mathcal{D}$.

## 4 Experiments

**Setup.** We model mesh vertices as graph nodes and establish edges according to the mesh topology, using additional edges between different objects based on Euclidean distance. We employ one-hot encoding to distinguish deformable objects from colliders, using a homogeneous graph representation (Pfaff et al., 2021). The edges additionally encode the relative distances between their connected nodes. Compared to existing work (Linkerhägner et al., 2023), we include a small initial context sequence covering time steps $1, \dots, T^c$ for each simulation, with $T^c$ being the anchor time step. Both the context and simulator MPNs use 15 message passing steps. Each message passing step uses separate 1-layer MLPs with a latent dimension of 128 and LeakyReLU activations for its node and edge updates.

We evaluate the *Full-rollout MSE*, computed as the average simulation MSEs following all time steps after the anchor time step, and the *Per-timestep MSE*, defined as the MSE at each individual time step. Both metrics are averaged over all trajectories in the test set. For each experiment, we report the mean and bootstrapped confidence intervals (Agarwal et al., 2021) over 8 random seeds. We evaluate both metrics for various context sizes ranging from 2 to 30 steps. Appendix C provides additional details on our experimental setup.

**Datasets.** We validate our method on five different simulation datasets based on three different mesh-based physics simulators. These include a *Deformable Block* (DB) task in 2D and a 3D *Tissue Manipulation* (TM) task (Linkerhägner et al., 2023), generated using Simulation Open Framework Architecture (SOFA) (Faure et al., 2012). In both datasets, the Poisson's ratio (Lim, 2015) acts as the randomized material property.

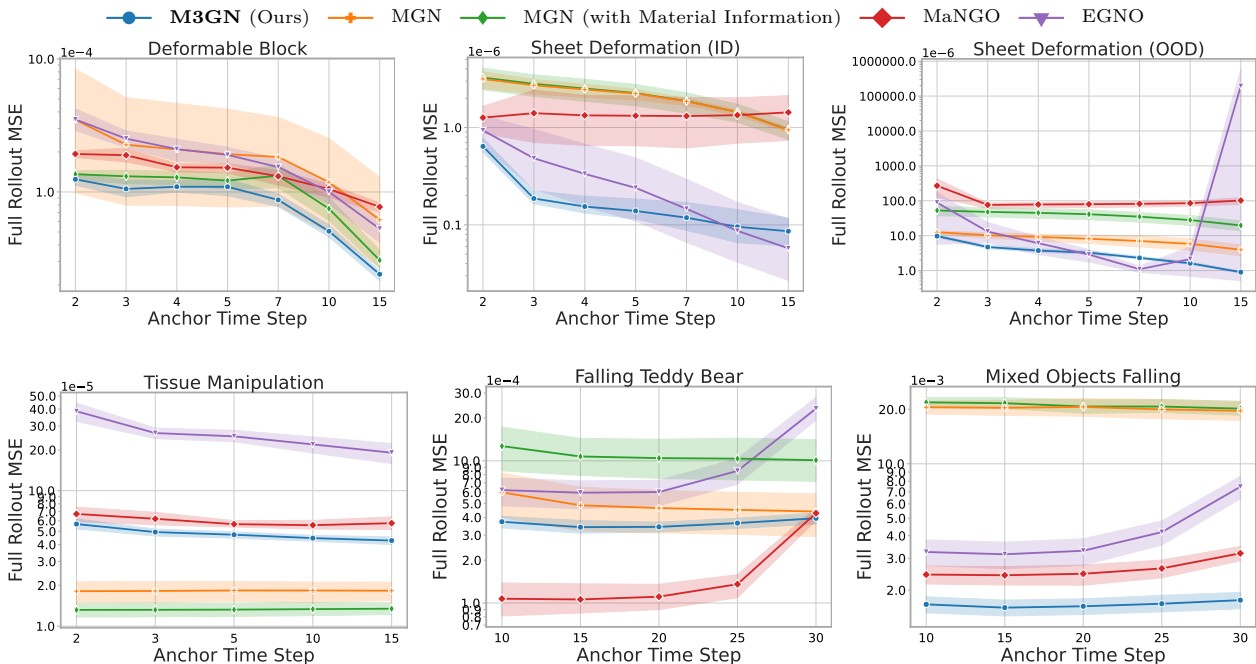

Figure 5: MSE (on a log scale) over full rollouts for different methods across all tasks, including an additional plot for the *Sheet Deformation* task evaluated on an out-of-distribution (OOD) test set of material properties. Overall, M3GN steadily improves its performance when provided with additional context information. Our method generally outperforms both MGN variants, likely due to the MP-based trajectory formulation and the latent material property representation, with the exception of the Tissue Manipulation task. Equivariant Graph Neural Operator (EGNO) exhibits instability for later anchor time steps and only performs competitively on the *Sheet Deformation (ID)* task.

*Deformable Block* simulates different trapezoids that are deformed by a circular collider with constant velocity and varying size and starting position. Each trajectory consists of a mesh with 81 nodes that is deformed over 39 time steps. *Tissue Manipulation* considers a surgical robotics scenario where a piece of tissue is deformed by a gripper. The gripper is attached to a fixed object position and moves in a random direction with constant velocity. The mesh comprises 361 nodes and the simulation has 100 steps.

We further consider the *Sheet Deformation* (SD) dataset (Dahlinger et al., 2025), which simulates how a sheet deforms under two constant forces that are applied at different positions of the sheet plane. As the Young's modulus is varied between sheets, this dataset constitutes a simplified stamp forming process, as common in mechanical engineering (Zimmerling et al., 2022). The simulations are generated with Abaqus (Smith, 2009), comprising 50 time steps and a plate with 225 nodes. We test two different data splits: The in-distribution (ID) split tests on material properties that the model has seen during training, while the out-of-distribution (OOD) split evaluates Young's modulus values outside the training domain.

The final two tasks place a randomly rotated deformable object at a specific height and let it fall to and collide with the ground. *Falling Teddy Bear* (FTB) considers the titular teddy bear as its only object, whereas six different objects are considered for *Mixed Objects Falling* (MOF). Each trajectory assigns a random Poisson's ratio and random Young's modulus to the falling object, thus influencing its deformation upon contact with the floor. Each trajectory in the last two tasks consists of 200 time steps. The object meshes have up to 350 nodes and are shown in Figure 11 in the appendix. For simplicity, we only consider the triangular surface meshes for the experimental setup. All task spaces are normalized to $[-1, 1]^3$. Appendix B.1 details the graph encoding, while Appendix D provides further information on dataset sizes and preprocessing.

**Baselines and Ablations.** We compare to MGN(Pfaff et al., 2021), evaluating its performance both with and without additional *material information* provided as a node feature. Importantly, we never supply this feature to M3GN.

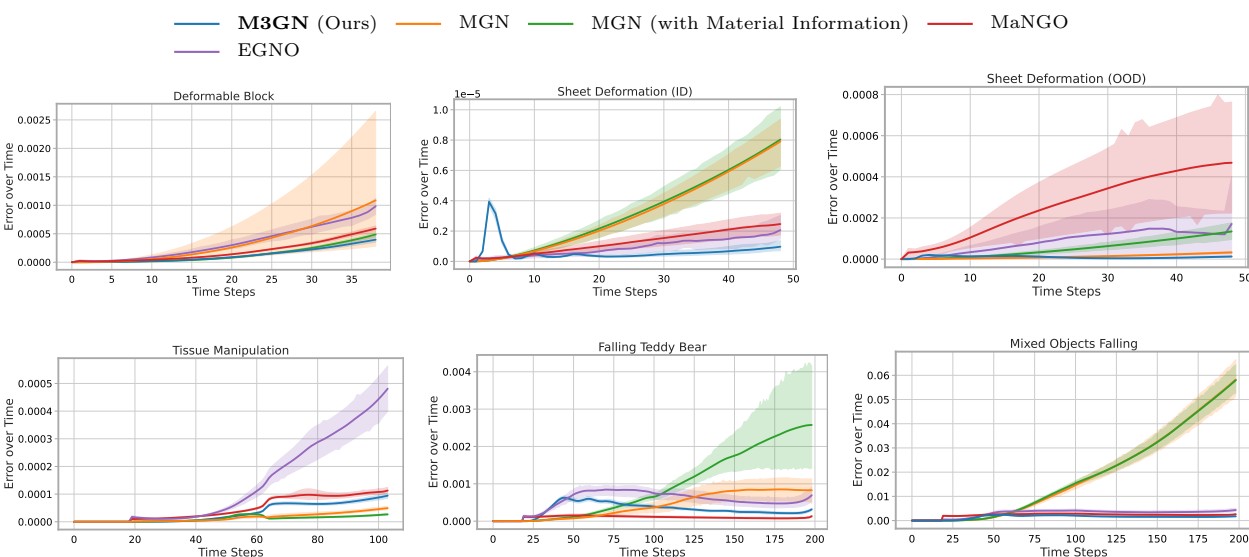

Figure 6: *Per-timestep MSE* for different methods across all tasks, showing the temporal evolution of prediction error averaged over all test trajectories. For DB, SD (ID), and SD (OOD), the context size is set to 2, whereas TM, FTB, and MOF use a context size of 20. The step-based MGN variants exhibit clear error accumulation over time, whereas M3GN maintains stable accuracy throughout the rollout likely due to its temporally consistent MP-based trajectory formulation.

MGN generates the next mesh state by iteratively predicting the velocities for the current simulation step. It is trained to minimize the 1-step MSE over node velocities, incorporating Gaussian input noise during training (Brandstetter et al., 2022). This noise serves to mitigate error accumulation and stabilize auto-regressive rollouts during inference. To ensure a fair comparison, we adopt the same hyperparameters as our method and optimize the input noise level per task.

We also explore the effect of incorporating historical information, specifically previous velocities, as node features for both MGN and M3GN. For MGN, using both the current and previous velocities improves performance significantly on many tasks. For M3GN, including only the current velocity yields similar benefits. Figure 12 in the Appendix presents the results of preliminary tuning experiments on a validation split. Additionally, Table 1 summarizes the specific history configurations and other relevant hyperparameter for each method and task. As a comparison to a graph-based meta-learning method, we evaluate M3GN against the Meta Neural Graph Operator (MaNGO) (Dahlinger et al., 2025). To adapt MaNGO to our trajectory-based meta-learning setup, we modify its architecture accordingly. Specifically, we remove the MaNGO encoder, which aggregates information across multiple trajectories. Instead, we provide the context data as the initial position in the unprocessed input trajectory of the MaNGO decoder and repeat the anchor-timestep data until the end of the trajectory. MaNGO then processes this sequence using spatial message passing and temporal convolution to produce its predicted trajectory. We discard the predicted portion corresponding to the context time steps and use the remaining time steps as the MaNGO prediction.

As an additional baseline, we compare our approach to the Equivariant Graph Neural Operator (EGNO) (Xu et al., 2024), which employs equivariant message-passing layers and predicts the remaining simulation steps in a single pass, closely aligning with our setup. However, training EGNO proved unstable with 15 message-passing steps, and the best results were achieved using only 5 steps. We hypothesize that this instability may stem from the longer prediction horizon of up to 200 steps in our experiments, as the baseline was originally evaluated on tasks with a much shorter prediction horizon of only 8 steps. Further details on the implementation of these baselines can be found in Appendix C.

We further study different design choices of M3GN on *Sheet Deformation* and *Deformable Block*. To investigate the effect of the meta-learning approach, we train an MGN (MP) variant that uses ProDMPs predictions, but omits a context aggregation and thus has no latent task description $z_v$. Similarly, we compare to

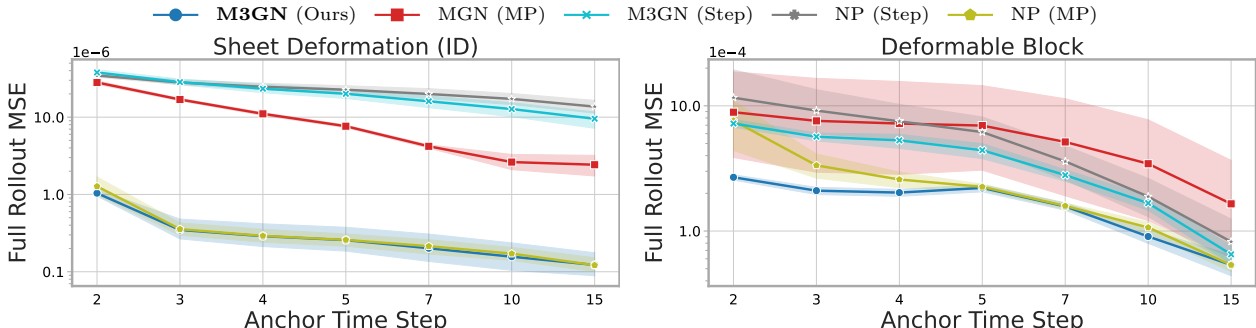

Figure 7: MSE on a log scale over full rollouts for the *Sheet Deformation* (**Left**) and *Deformable Block* (**Right**) tasks for different meta-learning and MP variants. Using a ProDMP representation for MGN improves performance. CNPs and NPs with a next-step prediction do not improve over standard MGN. A NP instead of an CNP architecture for M3GN slightly reduces performance.

M3GN (Step-based), which performs a next-step prediction of the dynamics instead of predicting ProDMP parameters, but otherwise follows the CNP training scheme to learn a latent task description. Finally, we compare the deterministic CNP approach to both MP and step-based probabilistic Neural Process (NP) approaches. Here, we get diagonal Gaussian distributions as the outputs of the context MPN, which we aggregate using Bayesian context aggregation (Volpp et al., 2021). We further investigate if node aggregation of the latent task description is beneficial by applying a maximum aggregation of the node features before the context aggregation. While standard CNPs require a permutation-invariant context aggregation, our context has a temporal structure. We thus experiment with a small transformer model with 4 transformer blocks, 4 attention heads, temporal encoding and a latent dimension of 32 as an aggregator. The transformer takes the sequence of outputs of the context MPN and predicts the aggregated node-level task description $z_v$.

**Main Results.** Figure 4 visualizes exemplary final simulation steps for M3GN and MGN for all tasks. M3GN aggregates context information to condition node-level ProDMP representations of the simulated trajectory. This approach leads to accurate simulations, providing much better alignment to the ground truth simulation than the step-based MGN on all tasks. Appendix E.4 shows visualizations of full simulation rollouts for all tasks and methods.[3]

Figure 5 provides the full-rollout MSE for M3GN, MGN, and MGN (with Material Information) across tasks. Adding material information improves performance only on the *Deformable Block* and *Tissue Manipulation* datasets. In contrast, on the *Falling Teddy Bear* and *Sheet Deformation (OOD)* tasks, MGN (with Material Information) tends to overfit, likely resulting in decreased performance when material information was provided. In general, using a later anchor time step improves performance across all methods, presumably due to a shorter prediction horizon. The main exception is EGNO on *Sheet Deformation (OOD)*, which becomes unstable for later anchor time steps.

M3GN surpasses all baselines *Deformable Block*. For *Sheet Deformation*, M3GN and EGNO significantly outperform the other baselines across context sizes. Furthermore, M3GN generalizes well to unseen material properties in the *OOD* setup, whereas, e.g., MGN and MaNGO fail to properly extrapolate. For *Tissue Manipulation*, the step-based baselines slightly outperform M3GN in terms of MSE. Yet, M3GN and MaNGO provide plausible and visually consistent simulations, with the step-based baselines primarily capturing finer details more accurately. In contrast, EGNO performs significantly worse in this setting, achieving an MSE that is an order of magnitude higher than that of M3GN.

On *Falling Teddy Bear* and *Mixed Objects Falling*, the step-based MGN without material information performs well on the former but, along with the other step-based method, fails to provide accurate long-term simulations on the latter. For *Mixed Objects Falling*, the predicted trajectories of both MGN variants qualitatively deviate from the ground truth. Both MGN and MGN (with Material Information) exhibit object drift or misalignment, e.g., between colliding bodies. In contrast, for M3GN, the ProDMP's temporally consistent movements

---

[3]Videos of these simulations are in the supplementary material.

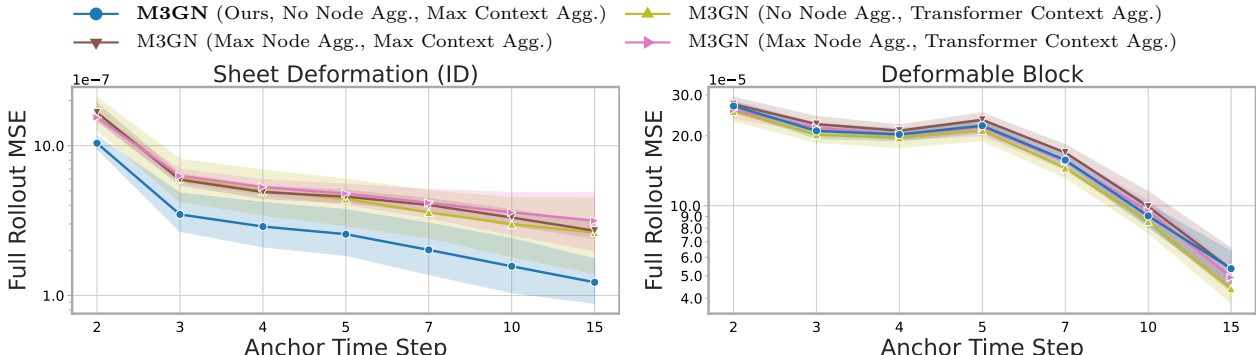

Figure 8: MSE on a log scale over full rollouts for the *Sheet Deformation* (**Left**) and *Deformable Block* (**Right**) tasks for different context and node aggregation methods. The node-level maximum context aggregation of M3GN performs best for *Sheet Deformation*, while all methods work roughly equally well on the *Deformable Block* task.

combined with context aggregation produce stable and coherent simulations, substantially improving over the step-based baselines. Examples of these behaviors are shown at the bottom of Figure 4. MaNGO achieves exceptional good results on *Falling Teddy Bear*, but it is surpassed by M3GN on *Mixed Objects Falling*. Interestingly, trajectory-based methods do not benefit from later anchor steps on these tasks. We suspect the increase in MSE is a consequence of how the metric is averaged: early timesteps are trivial (the object is simply falling), so including them lowers the mean error. When anchoring later, these easy steps are removed, and the MSE is computed only over the harder, post-impact segment, leading to a higher average error.

**Additional Experiments.** Figure 6 shows representative *Per-timestep MSE* results, while Appendix E provides the complete set of evaluations for different context sizes. These figures highlight the error accumulation common in step-based learned simulatiors. In contrast, M3GN maintains stable accuracy throughout the rollout. In *Sheet Deformation (ID)*, we observe a temporary increase in error for M3GN around time steps 2 to 5. This behavior seems to stem from a rapid change in node velocities, which our ProDMP representation smooths over. While additional basis functions would increase the ProDMP's capacity and thus likely mitigate this issue, we deliberately avoided excessive hyperparameter tuning to ensure a fair comparison and comparable optimization budgets across methods.

Next, Figure 7 finds that both MP representations and a meta-learning objective are crucial for accurate simulations. Interestingly, using NPs with ProDMPs only slightly degrades performance compared to M3GN, suggesting that the NP's latent distribution does not benefit our GNS setup. Figure 8 additionally compares different methods to aggregate the context set. The node-level maximum context aggregation works best for *Sheet Deformation*. For *Deformable Block*, there is no significant difference between aggregations, causing us to use the comparatively simple node-level maximum context aggregation for all experiments.

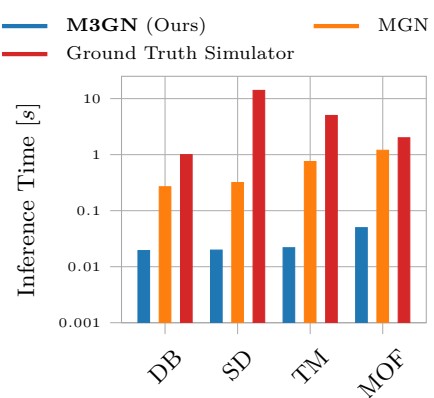

Figure 9: Runtime comparison on four tasks between the learned methods and the different ground truth simulators. Note the log scale on the y-axis.The complete runtime and memory usage evaluation is in Appendix, Table 3.

Figure 19 in the Appendix presents two additional ablation studies that further characterize the behavior of M3GN. We first investigate the stability of M3GN under noisy context observations and compare it against MGN as well as a variant of M3GN that does not receive any context data. While M3GN remains robust under moderate noise levels, we observe that the highest noise setting leads to worse performance than the no-context variant, which we attribute to overfitting to uninformative and misleading context signals. We further evaluate generalization across materials using a dataset composed of five distinct material parameter

settings by comparing it to specialized baselines. Five separate MGN models are trained, each specialized to a single material and evaluated on unseen initial conditions with the same material properties, serving as an upper bound for single-material approaches. In contrast, a single M3GN model is trained on the combined dataset and must infer material properties from the context set. For small context sizes, M3GN achieves performance comparable to the specialized MGN models, while for larger context sizes M3GN slightly outperforms MGN, demonstrating in total its ability to effectively leverage context information and generalize across materials within a unified model.

We additionally present a visualization of the latent space for M3GN in Figure 20 in the Appendix, showing that simulations with similar material properties are clustered together. This latent structure highlights the model's ability to differentiate between material behaviors while preserving the relationships between similar properties.

Figure 9 compares the inference speed of M3GN to that of the step-based MGN. M3GN's ProDMP trajectory representation significantly decreases the amount of required model calls for the simulation, since, after encoding, a single GNN forward pass can is used to compute the full trajectory. Additionally, the context set is easily encoded in parallel, resulting in a relatively minor cost for the context computation and aggregation for all tasks. While MGN does not perform any context processing, it requires one forward pass per timestep, making its rollout inherently sequential. Together, these properties result in an inference-time speedup of up to a factor of 32 for M3GN compared to MGN (for the *Tissue Manipulation* task), and up to a factor of 700 compared to the ground-truth simulators (for the *Sheet Deformation* task). A complete comparison of execution time and memory consumption during inference is provided in Table 3 in the Appendix. Improving over existing learned simulators by more than one, and over classical simulators by more than two orders of magnitude enables rapid development for downstream engineering applications, such as manufacturing optimization.

## 5 Conclusion

We introduce Movement-Primitive Meta-MeshGraphNet (M3GN), a novel Graph Network Simulator that combines movement primitives with trajectory-level meta-learning for efficient and accurate long-term predictions in physical simulations. Our method dynamically adapts to available context information during inference, enabling accurate prediction of deformations under unknown object properties. Additionally, it effectively mitigates error accumulation while reducing the number of required learned simulator function calls. To validate the effectiveness of M3GN, we propose two new deformation prediction tasks with uncertain material properties. Results on these tasks and existing datasets demonstrate that our method consistently outperforms two strong Graph Network Simulator baselines, even when the baselines are provided with oracle information about the material properties.

**Limitations and Future Work.** Our method may struggle to capture rapid, high-frequency dynamics, as the smooth ProDMP formulation can lead to temporary errors when node velocities change abruptly. We also plan to integrate online re-planning of trajectories, predicting trajectory segments with every model forward pass. This process may enhance coordination between simulated nodes across segments while maintaining the benefits of a compact multi-step trajectory representation. In addition, the M3GN framework is not limited to deformation tasks; it can be extended to new settings such as fluid simulations or other non-mesh-based environments, provided that an appropriate graph structure can be constructed from the underlying data.

**Broader Impact Statement** Our proposed Graph Network Simulator can positively impact various fields relying on computational modeling and simulation by significantly reducing computational cost compared to traditional simulators while providing accurate simulations. However, efficient and accurate simulation of physical systems also comes with potential negative impacts, such as, e.g., the development of advanced weapon models.

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

# A   Mathematical formulations of Movement Primitives

We provide an overview of the probabilistic dynamic movement primitives (ProDMP) formulations utilized in this paper, starting with the foundational methods: Dynamic Movement Primitives (DMPs) and Probabilistic Movement Primitives (ProMPs).

## A.1   DMPs

Schaal (2006) introduced Dynamic Movement Primitives (DMPs), which integrate a forcing term into a dynamical system to generate smooth trajectories from given initial conditions[4], such as a robot's position and velocity at a particular time. A DMP trajectory is governed by a second-order linear ordinary differential equation (ODE) as follows:

$$\tau^2 \ddot{y} = \alpha(\beta(g - y) - \tau \dot{y}) + f(x), \quad f(x) = x \frac{\sum \varphi_i(x) w_i}{\sum \varphi_i(x)} = x \boldsymbol{\varphi}_x^\mathsf{T} \boldsymbol{w}, \tag{7}$$

where $y = y(t)$, $\dot{y} = \mathrm{d}y/\mathrm{d}t$, and $\ddot{y} = \mathrm{d}^2 y/\mathrm{d}t^2$ denote the position, velocity, and acceleration of the system at a specific time $t$, respectively. Constants $\alpha$ and $\beta$ are spring-damper parameters, $g$ is the goal attractor, and $\tau$ is a time constant modulating the speed of trajectory execution.

The functions $\varphi_i(x)$ represent the basis functions for the forcing term, as shown in Fig. 10a, while the phase variable $x = x(t) \in [0, 1]$ captures the execution progress. The trajectory's shape is determined by the weight parameters $w_i \in \boldsymbol{w}$ for $i = 1, \ldots, N$ and the goal term $g$. The trajectory $[y_t]_{t=0:T}$ is typically computed by numerically integrating the dynamical system from the start to the endpoint. However, this numerical process is computationally expensive (Bahl et al., 2020; Li et al., 2023), as its cost scales with the trajectory length and the resolution of the numerical integration.

## A.2   ProMPs

Paraschos et al. (2013) introduced the Probabilistic Movement Primitives (ProMPs) framework for modeling trajectory distributions, effectively capturing both temporal and inter-dimensional correlations. Unlike DMPs, which rely on a forcing term, ProMPs directly model the desired trajectory and its distribution using a linear basis function representation. Given a weight vector $\boldsymbol{w}$ or a weight vector distribution $p(\boldsymbol{w}) \sim \mathcal{N}(\boldsymbol{w}|\boldsymbol{\mu_w}, \boldsymbol{\Sigma_w})$, the corresponding trajectory or trajectory distribution is computed as follows:

$$\text{Compute Trajectory:} \quad [y_t]_{t=0:T} = \boldsymbol{\Phi}^\mathsf{T} \boldsymbol{w}, \tag{8}$$

$$\text{Compute Distribution:} \quad p([y_t]_{t=0:T}; \; \boldsymbol{\mu_y}, \boldsymbol{\Sigma_y}) = \mathcal{N}(\boldsymbol{\Phi}^\mathsf{T} \boldsymbol{\mu_w}, \; \boldsymbol{\Phi}^\mathsf{T} \boldsymbol{\Sigma_w} \boldsymbol{\Phi}). \tag{9}$$

Here, the matrix $\boldsymbol{\Phi}$ contains the basis functions for each time step $t \in [0, T]$, shown in Fig. 10a. The trajectory shape is determined by the weight parameters $w_i \in \boldsymbol{w}$ through matrix-vector multiplication. Despite their simplicity and computational efficiency, ProMPs lack an intrinsic dynamic system, limiting their ability to specify a given initial condition for a trajectory or predict smooth transitions between two ProMP trajectories with differing parameter vectors.

## A.3   ProDMPs

**Solving the ODE underlying DMPs**   Li et al. (2023) observed that the governing equation of DMPs, as described in Eq. (7), admits an analytical solution. We re-express the original ODE from Eq. (7) and its homogeneous counterpart in standard ODE forms as follows:

$$\text{Non-homo. ODE:} \quad \ddot{y} + \frac{\alpha}{\tau}\dot{y} + \frac{\alpha\beta}{\tau^2}y = \frac{f(x)}{\tau^2} + \frac{\alpha\beta}{\tau^2}g \equiv F(x, g), \tag{10}$$

$$\text{Homo. ODE:} \quad \ddot{y} + \frac{\alpha}{\tau}\dot{y} + \frac{\alpha\beta}{\tau^2}y = 0. \tag{11}$$

---

[4]In mathematics, an initial condition refers to the value of a function or its derivatives at a starting point, which can be specified at any time, not necessarily at $t = 0$.

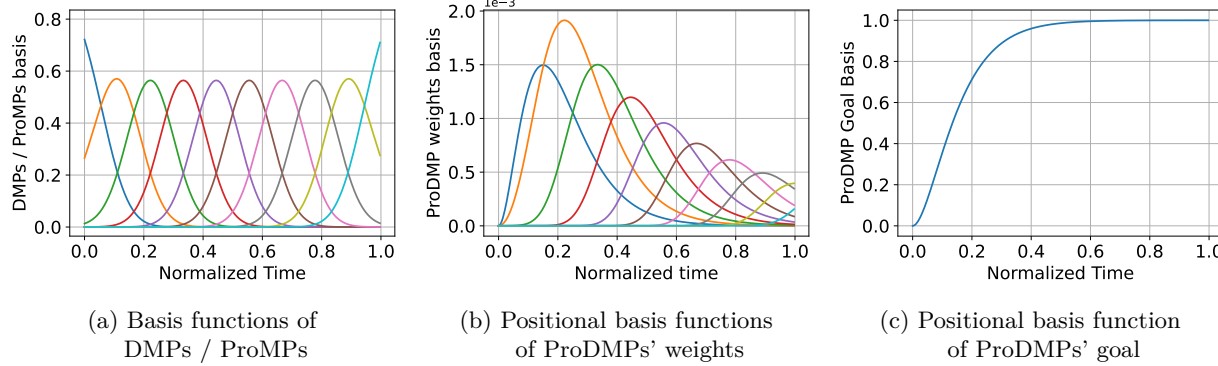

(a) Basis functions of
DMPs / ProMPs

(b) Positional basis functions
of ProDMPs' weights

(c) Positional basis function
of ProDMPs' goal

Figure 10: Illustration of basis functions used in MP methods. (a) Normalized radial basis functions used in DMPs in Eq.(7) and ProMPs in Eq.(8), respectively. (b) Positional basis functions of ProDMPs' weights $\boldsymbol{w}$ and (c) ProDMPs' goal $g$ in Eq.(17). In ProDMPs, $g$ is concatenated with the weights vector $\boldsymbol{w}$ and treated as one dimension of the resulting vector $\boldsymbol{w}_g$. Both weights and goal basis functions are computed from solving the DMPs' underlying ODE, following the procedure from Eq.(12) to Eq.(16)

The solution to this ODE is essentially the position trajectory, and its time derivative yields the velocity trajectory. They are formulated through several time-dependent function as:

$$y = \begin{bmatrix} y_2\boldsymbol{p_2} - y_1\boldsymbol{p_1} & y_2q_2 - y_1q_1 \end{bmatrix} \begin{bmatrix} \boldsymbol{w} \\ g \end{bmatrix} + c_1y_1 + c_2y_2 \tag{12}$$

$$\dot{y} = \begin{bmatrix} \dot{y}_2\boldsymbol{p_2} - \dot{y}_1\boldsymbol{p_1} & \dot{y}_2q_2 - \dot{y}_1q_1 \end{bmatrix} \begin{bmatrix} \boldsymbol{w} \\ g \end{bmatrix} + c_1\dot{y}_1 + c_2\dot{y}_2. \tag{13}$$

Here, the learnable parameters $[\boldsymbol{w}, g]^T$ which control the shape of the trajectory, are separable from the remaining time-dependent functions $y_1, y_2, \boldsymbol{p}_1, \boldsymbol{p}_2, q_1, q_2$. These functions are computed by solving the ODE in Eq. (10), (11):

$$y_1(t) = \exp\left(-\frac{\alpha}{2\tau}t\right), \qquad\qquad y_2(t) = t\exp\left(-\frac{\alpha}{2\tau}t\right), \tag{14}$$

$$\boldsymbol{p}_1(t) = \frac{1}{\tau^2}\int_0^t t'\exp\left(\frac{\alpha}{2\tau}t'\right)x(t')\boldsymbol{\varphi}_x^\intercal \mathrm{d}t', \qquad \boldsymbol{p}_2(t) = \frac{1}{\tau^2}\int_0^t \exp\left(\frac{\alpha}{2\tau}t'\right)x(t')\boldsymbol{\varphi}_x^\intercal \mathrm{d}t', \tag{15}$$

$$q_1(t) = \left(\frac{\alpha}{2\tau}t - 1\right)\exp\left(\frac{\alpha}{2\tau}t\right) + 1, \qquad q_2(t) = \frac{\alpha}{2\tau}\left[\exp\left(\frac{\alpha}{2\tau}t\right) - 1\right]. \tag{16}$$

Here, the function $y_1, y_2$ are the complementary solutions to the homogeneous ODE presented in Eq.(11), with $\dot{y}_1, \dot{y}_2$ their time derivatives respectively.

It's worth noting that $\boldsymbol{p}_1$ and $\boldsymbol{p}_2$ cannot be derived analytically due to the complexity of the forcing basis terms $\boldsymbol{\varphi}_x$. Consequently, these terms must be computed numerically. However, isolating the learnable parameters, namely $\boldsymbol{w}$ and $g$, enables the reuse of other time-dependent functions across all generated trajectories.

ProDMPs identify these reusable terms as the position and velocity basis functions, denoted by $\boldsymbol{\Phi}(t)$ and $\dot{\boldsymbol{\Phi}}(t)$, respectively. Fig. 10b and Fig. 10c illustrate the resulting position basis functions for the weights $\boldsymbol{w}$ and the goal $g$, respectively. These functions are pre-computed offline and treated as constants during online learning. When $\boldsymbol{w}$ and $g$ are combined into a concatenated vector, represented as $\boldsymbol{w}_g$, the position and velocity trajectories can be expressed in a manner similar to that used by ProMPs:

$$\textbf{Position:} \quad y(t) = \boldsymbol{\Phi}(t)^\intercal \boldsymbol{w}_g + c_1y_1(t) + c_2y_2(t), \tag{17}$$

$$\textbf{Velocity:} \quad \dot{y}(t) = \dot{\boldsymbol{\Phi}}(t)^\intercal \boldsymbol{w}_g + c_1\dot{y}_1(t) + c_2\dot{y}_2(t). \tag{18}$$

In the main paper, for simplicity and notation convenience, we use $\boldsymbol{w}$ instead of $\boldsymbol{w}_g$ to describe the parameters and goal of ProDMPs.

**Trajectory's Initial Condition**   The coefficients $c_1$ and $c_2$ are solutions to the initial value problem defined by Eqs.(17)(18). Assuming the trajectory starts at time $t_b$ with position $y_b$ and velocity $\dot{y}_b$, we denote the values of the complementary functions and their derivatives at the condition time $t_b$ as $y_{1_b}, y_{2_b}, \dot{y}_{1_b}$ and $\dot{y}_{2_b}$. Similarly, the values of the position and velocity basis functions at $t_b$ are denoted as $\boldsymbol{\Phi}_b$ and $\dot{\boldsymbol{\Phi}}_b$ respectively. Using these notations, $c_1$ and $c_2$ are computed as:

$$
\begin{bmatrix} c_1 \\ c_2 \end{bmatrix} = \begin{bmatrix} \frac{\dot{y}_{2_b} y_b - y_{2_b} \dot{y}_b}{y_{1_b} \dot{y}_{2_b} - y_{2_b} \dot{y}_{1_b}} + \frac{y_{2_b} \dot{\boldsymbol{\Phi}}_b^\mathsf{T} - \dot{y}_{2_b} \boldsymbol{\Phi}_b^\mathsf{T}}{y_{1_b} \dot{y}_{2_b} - y_{2_b} \dot{y}_{1_b}} \boldsymbol{w}_g \\ \frac{y_{1_b} \dot{y}_b - \dot{y}_{1_b} y_b}{y_{1_b} \dot{y}_{2_b} - y_{2_b} \dot{y}_{1_b}} + \frac{\dot{y}_{1_b} \boldsymbol{\Phi}_b^\mathsf{T} - y_{1_b} \dot{\boldsymbol{\Phi}}_b^\mathsf{T}}{y_{1_b} \dot{y}_{2_b} - y_{2_b} \dot{y}_{1_b}} \boldsymbol{w}_g \end{bmatrix}. \tag{19}
$$

**Set Goal Convergence Relative to Initial Condition**   The goal attractor $g$ in the ProDMPs framework represents an asymptotic convergence point for the dynamical system as $t \to \infty$, typically defined as an absolute coordinate. However, the goal term can also be modeled relative to the initial position $y_b$. In this approach, the relative goal $g_{\text{rel}}$ is predicted, and its absolute counterpart is computed as $g_{\text{abs}} = g_{\text{rel}} + y_b$. This approach is particularly useful for predicting the goal in the coordinate system relative to a node's starting position. Since we aim to achieve a translation-equivariant approach (where absolute node positions are encoded as relative edge features between nodes), predicting relative goal positions aligns well with this design principle.

# B   Architecture and Method Details

This section offers detailed insights into our methodology and the architectural decisions guiding our approach.

## B.1   Graph Encodings

In processing the initial graph $\mathcal{G}_{*,T^c}$, we create edges between the mesh and the collider based on a radius graph. Specifically, we connect mesh and collider nodes for *Deformable Block* and *Tissue Manipulation* if their euclidean distance is smaller than 0.3. In the *Tissue Manipulation* task, the collider is given as a single node which is connected to the tip of the tissue. It marks the grasping point of a gripper. In the *Sheet Deformation* task, we add an additional node feature to the nodes which get directly influenced by the external force. Therefore, no collider is used in this task. For the *Falling Teddy Bear* and the *Mixed Objects Falling* task, we implicitly model the ground as a collider by adding the current $z$ position of every node to its node features (Sanchez-Gonzalez et al., 2018). This quantity gets updated for the step-based methods.

## B.2   ProDMP Details

Initialization of ProDMPs necessitates node velocities for the anchor time step $T^c$. We employ a linear approximation, leveraging data from the previous time step $T^c - 1$.

Similar to the relative encoding of node positions in the MPN, we employ a technique in ProDMP to derive relative trajectories. Initially, we integrate a relative goal position as part of the node weights $\boldsymbol{w}_v$. Utilizing this approach, trajectories commence from the origin and traverse towards their respective relative goals. Subsequently, we adjust all positions by the initial position. This strategy fosters model generalization across various nodes.

The parameter $\tau$, as described in Equation 7, is learned globally across all tasks using a compact MLP. The model's final layer employs a scaled sigmoid function for parameter estimation.

# C   Experimental Protocol

In order to promote reproducibility, we provide details of our experimental methodology. Table 1 presents the hyperparameters used in our experiments. For a comprehensive description of the creation of all datasets, please refer to Appendix D.

The training took place on an NVIDIA A100 GPU, with each method given the same computation budget of 48 hours, except for the *Sheet Deformation* task, where the computation budget was set to 24 hours. Consequently, the number of epochs varied, as the batching differed significantly between the trajectory-based method M3GN and the step-based MGN. We adapted the batchsize of the step-based methods in order to use the GPU memory efficiently. M3GN is always trained on one full trajectory per batch. Here, the whole context is processed in parallel and the remaining trajectory is predicted and compared to the ground truth.

We conducted a multi-staged grid-based hyperparameter search for the learning rate, input noise, and other hyperparameters as the latent task description dimension. In general, we optimized all methods on all tasks separately, however, we noticed that over different tasks and methods some parameters had the same best configuration. We did not use the test data for this, but tuned all hyperparameters on a separate validation split. This split was also used to determine the best epoch checkpoint to mitigate any overfitting effects.

In the end, all methods worked well with a learning rate of $5.0 \times 10^{-4}$ except in the *Sheet Deformation* task. Here, our hyperparameter optimization indicated that the trajectory based methods benefit from a smaller learning rate of $1.0 \times 10^{-5}$. For MGN, we experimented with different input noise scales. Notably, for the *Deformable Block* and the *Sheet Deformation* task, a smaller noise scale improved performance significantly. In the falling objects tasks, we also explored second-order predictions, such as node accelerations, instead of velocity predictions. Following the approach in Pfaff et al. (2021), we adjusted the labels accordingly and conducted preliminary evaluations. However, since direct velocity predictions yielded superior results, we opted for them as our final approach, as presented in the main paper.

### C.1   EGNO Training

For the Equivariant Graph Neural Operator (EGNO) method, we used the original code from Xu et al. (2024) for the model implementation. Since EGNO can only predict for a fixed next horizon, we cut the remaining prediction when using a later anchor time step. This is done during training and evaluation.

### C.2   MGN Training

We mainly follow Pfaff et al. (2021) for the training of the MGN baseline. The only difference is the incorporation of current and historic velocity node features. Pfaff et al. (2021) consider this in their experiments but they show in their experiment suite that it does not improve the results and can lead to overfitting. This is different to our results. For us, on all tasks except *Mixed Objects Falling*, adding the current and historic velocities of nodes improves the results. We follow the Gaussian random walk noise injection for the velocity features from Sanchez-Gonzalez et al. (2020).

## D   Datasets and Preprocessing information

In this section, we give detailed information about the datasets we used. We report a general overview of all datasets in Table 2. Here each dataset is abbreviated for brevity as the following:

Table 2 lists in detail the datasets used in the paper. Each dataset is abbreviated for brevity and explained as follows:

- **SD.**: *Sheet Deformation*

- **DB**: *Deformable Block*

- **TM.**: *Tissue Manipulation*

- **FTB.**: *Falling Teddy Bear*

- **MOF.**: *Mixed Objects Falling*

Table 1: Table listing the hyperparameters and configurations of the experiments

| Parameter | Value |
|---|---|
| Node feature dimension | 128 |
| Latent task description dimension | 64 |
| Decoder hidden dimension | 128 |
| Message passing blocks | 15 |
| Message passing blocks (EGNO) | 5 |
| GNN Aggregation function | Mean |
| GNN Activation function | Leaky ReLU |
| M3GN Context Aggregation method | Max Aggr. |
| M3GN Latent Node Aggregation method | No Aggr. |
| Learning rate | $5.0 \times 10^{-4}$ |
| Learning rate (*Sheet Def.* MP-based methods) | $1.0 \times 10^{-5}$ |
| Number of ProDMP basis functions | 30 |
| ProDMP $\tau$ | learned range: $[0.3, 3.0]$ |
| ProDMP Relative start position | True |
| MGN input mesh noise | 0.01 |
| MGN input mesh noise (*Def. Block*) | 0.001 |
| MGN input mesh noise (*Sheet Def.*) | 0.0001 |
| MGN history length (all tasks except *Mixed Objects Fall*) | 2 |
| MGN history length (*Mixed Objects Fall*) | 0 |
| M3GN history length (all tasks except *Tiss. Man.* and *Sheet Def. (OOD)*) | 1 |
| M3GN history length (*Tiss. Man.* and *Sheet Def. (OOD)*) | 0 |
| Minimum/Maximum Train Context Size (*Sheet Deformation*) | 2 / 15 |
| Minimum/Maximum Train Context Size (*Deformable Block*) | 2 / 15 |
| Minimum/Maximum Train Context Size (*Tissue Manipulation*) | 2 / 40 |
| Minimum/Maximum Train Context Size (*Falling Teddy Bear*) | 10 / 50 |
| Minimum/Maximum Train Context Size (*Mixed Objects Fall*) | 10 / 50 |
| Threshold to create collider-mesh edge | 0.3 |

Table 2: Table listing the datasets and their configurations

| Name | Train/Val/Test Splits | Number of steps | Number of Nodes | Collider interaction |
|---|---|---|---|---|
| SD. | 630/135/135 | 50 | 225 | External Force |
| DB. | 675/135/135 | 39 | 81 | Rigid Collider |
| TM. | 600/120/120 | 100 | 361 | Grasping point |
| FTB. | 700/150/150 | 200 | 304 | Boundary Condition |
| MOF. | 1800/360/360 | 200 | up to 350 | Boundary Condition |

### D.1 Sheet deformation

We select 9 different Young's modulus ranging between 10 and 1000 from a very deformable to an almost stiff sheet. Then, per material, we compute 100 simulations using Abaqus where the positions of the two acting forces are randomized. The boundary nodes of the sheet are kept in place. From every material configuration, we take 70 simulations for training and 15 simulations for validation and testing respectively. For the out-of-distribution task, we trained using Young's modulus values ranging from 60 to 500 and tested using values of $10, 30, 750,$ and $1000$.

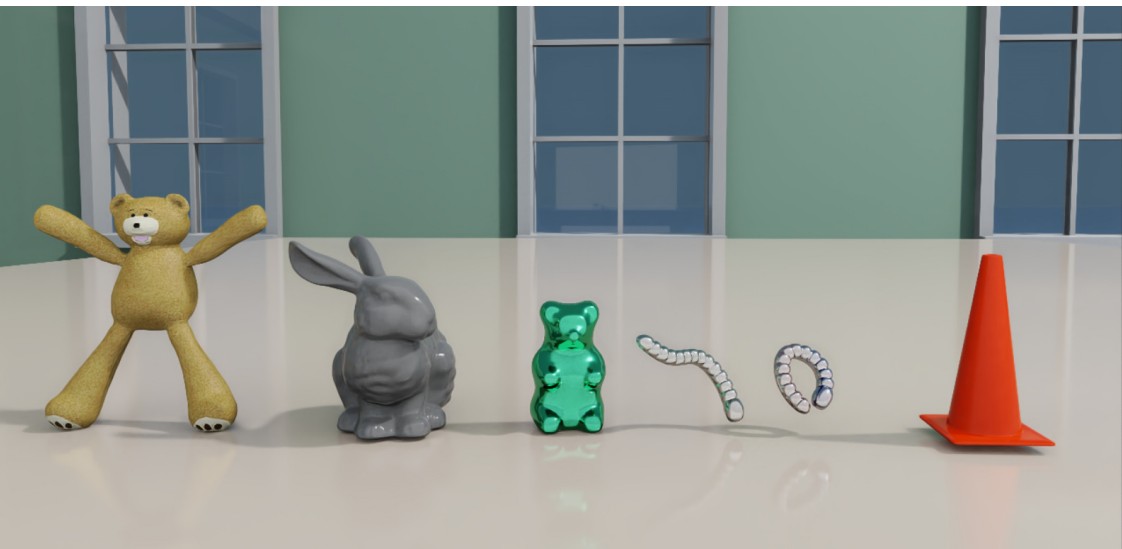

Figure 11: Six objects used in the *Mixed Objects Falling* task. From left to right: Teddy Bear, Bunny, Gummy Bear, Gummy Worm 1, Gummy Worm 2, and Traffic Cone.

### D.2    Deformable Block

The original task was introduced in Linkerhägner et al. (2023), generated using SOFA (Faure et al., 2012). It uses 3 different Poisson's ratios and 9 different trapezoidal meshes. We increase the difficulty of this dataset by introducing more complex initial starting conditions. This is done by selecting a random Poisson's ratio, simulating for 11 steps, and then switching to another Poisson's ratio. Then, the simulation continues for 39 steps. The first 11 steps are then discarded and step 12 is then the initial step for the dataset (and is referred to step 0 throughout the paper).

### D.3    Tissue Manipulation

We use the original task introduced in Linkerhägner et al. (2023) without alterations. This dataset was also generated using SOFA (Faure et al., 2012).

### D.4    Falling Teddy Bear

Each trajectory of the dataset was created by choosing an angle from $[0°, 360°]$ for the first time step. To vary the material properties, we randomly select combinations of Young's modulus and Poisson's ratio from 1000 uniformly spaced values of the Young's modulus in $[1 \times 10^5, 1 \times 10^6]$ and 100 uniformly spaced values of the Poisson's ratio in $[0.0, 0.499]$. This dataset is generated using NVIDIA Isaac Sim (NVIDIA, 2022a), which utilizes PhysX 5.0 (NVIDIA, 2022b) to simulate tetrahedral meshes based on initial CAD models.

### D.5    Mixed Objects Falling

The simulation uses the setup from *Falling Teddy Bear*. In addition to the Teddy, we include other objects to encourage diversity. In total, there are six different objects presented in the dataset. We report an image of their high-resolution meshes in Figure 11. From every object, we use 300 simulations for the training split and 60 simulations for the test and validation split respectively.

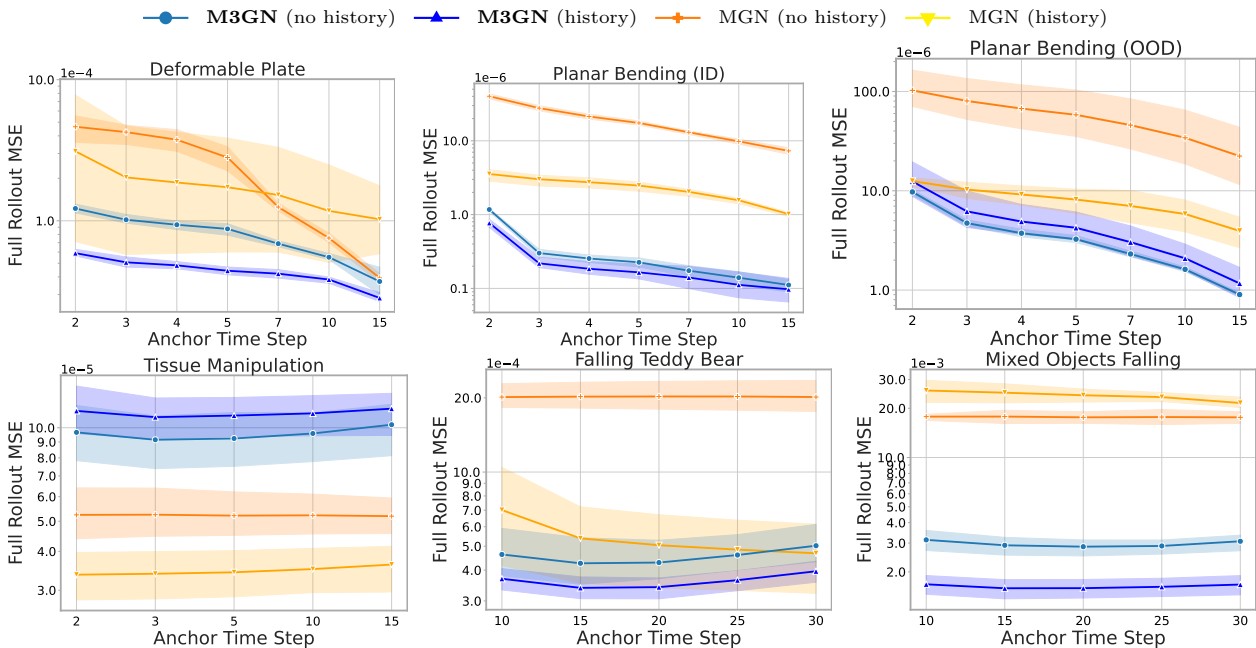

Figure 12: Log-scale MSE over full rollouts on the validation split for M3GN and MGN comparing history features. The better performing hyperparameter configuration was chosen for the final evaluation on the test dataset.

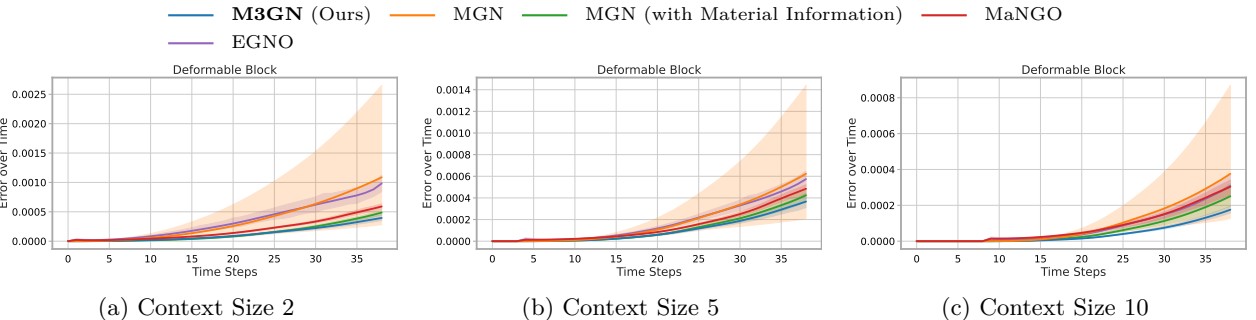

Figure 13: Per-timestep MSE for the *Deformable Block* task.

# E    Additional Results

## E.1    Hyperparameter Optimization

We observed that the history inclusion of previous velocities has a big impact on the result of the simulation, depending on the task. To obtain optimal performance, we did an hyperparameter optimization on the validation split comparing history features. The results for M3GN and MGN are given in Figure 12.

## E.2    MSE over time

To gain better insights into the rollout stability of the model predictions, we report the Mean Squared Error (MSE) over timesteps in Figure 13, 14, 15, 16, 17, and Figure 18. Overall, our model M3GN demonstrates great robustness against error accumulation, benefiting from the inherent trajectory representation provided by the ProDMP method. MGN works well on the *Tissue Manipulation* task, but fails to incorporate the correct context information on other tasks. EGNO performs in general worse except on the *Sheet Deformation* tasks.

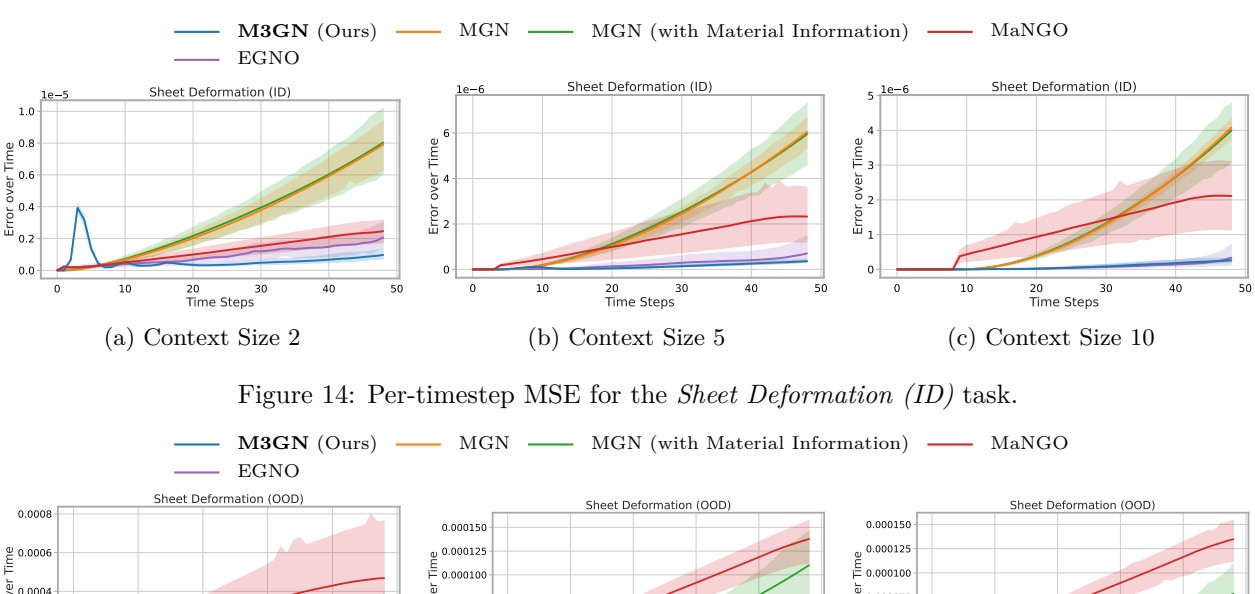

Figure 14: Per-timestep MSE for the *Sheet Deformation (ID)* task.

(a) Context Size 2     (b) Context Size 5     (c) Context Size 10

Figure 15: Per-timestep MSE for the *Sheet Deformation (OOD)* task.

### E.3  Additional Ablations

Figure 19 illustrates two complementary evaluations of M3GN. On the left, we analyze the stability of M3GN when provided with noisy context data and compare its behavior to MGN as well as to a variant of M3GN that operates without any context observations. This experiment highlights the robustness of M3GN to imperfect contextual information. On the right, we consider a dataset composed of five distinct material parameter settings. We train five separate MGN models, each specialized to a single material, and evaluate them on unseen initial conditions but with their assigned material properties. These models are therefore specialized to a single material and serve as an upper bound for single-material approaches. In contrast, we train one M3GN model on the union of all material splits, where the model must infer the underlying material properties directly from the context set, demonstrating its ability to generalize across materials within a unified model.

### E.4  Visualizations

In Figure 20, we include a latent space visualization of the *Sheet Deformation* task, where simulations with 9 different Young's Modulus values are clustered according to their material properties. The t-SNE projection of the 64-dimensional latent node vectors demonstrates clear clustering, indicating that the model effectively captures and differentiates material characteristics based on learned task representations.

We further provide additional visualizations for M3GN, MGN and MGN (with Material Information) for exemplary simulations of all tasks. Each visualization shows the same simulated trajectory for different time steps (columns) and different methods (rows).

- Figure 21 shows a simulation of the *Sheet Deformation* task for a context set size of 5.

- Figure 22 visualizes the *Deformable Block* task for a context set size of 6.

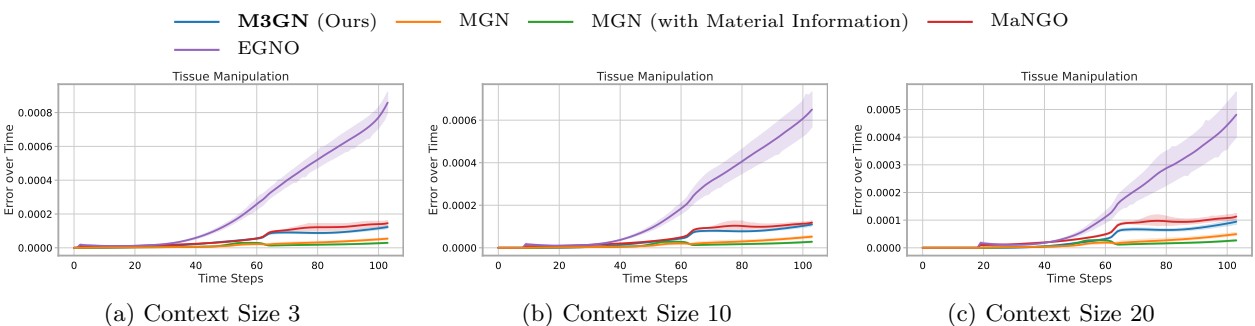

Figure 16: Per-timestep MSE for the *Tissue Manipulation* task.

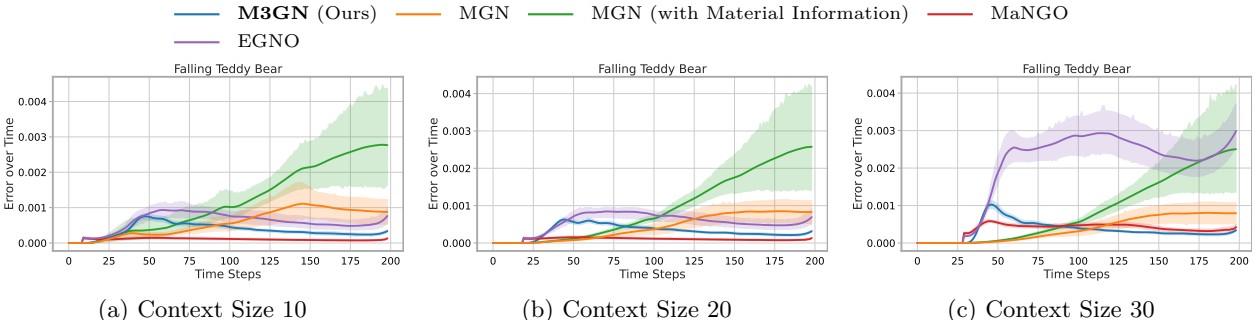

Figure 17: Per-timestep MSE for the *Falling Teddy Bear* task.

- Figure 23 shows a *Tissue Manipulation* visualization for a context set size of 6.

- Figure 24 provides an examplary *Teddy Bear Falling* for a context set size of 20.

- Figure 25 and Figure 26 show two different simulated *Mixed Objects Falling* for a context set size of 20.

Across tasks, M3GN provides accurate simulations, whereas MGN, especially when not provided the additional material information as oracle knowledge, sometimes fails to respect the material properties or predicts a drift in the solution for later time steps.

## E.5   Runtime, memory and parameter comparison.

In Table 3, we provide the runtime, memory, and parameter comparison of all methods on the four benchmark task during inference. We split up the total inference time into context encoding and simulation time for M3GN.

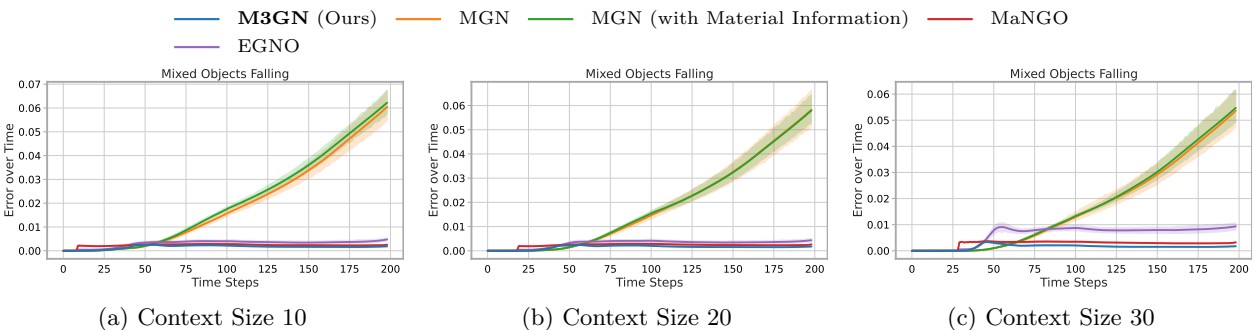

Figure 18: Per-timestep MSE for the *Mixed Objects Falling* task.

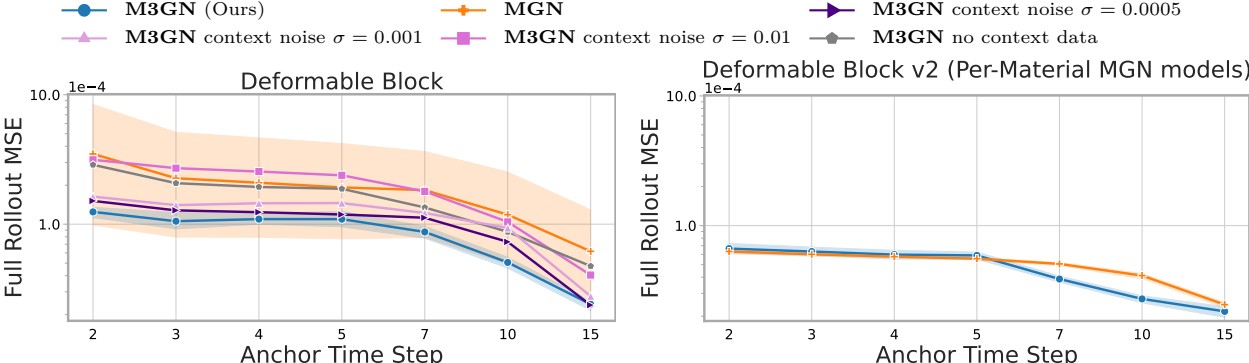

Figure 19: MSE on a log scale for two ablations on the Deformable Block Dataset. **Left:** Stability analysis of M3GN using noisy context data, compared with MGN and a variant of M3GN that does not observe any context data. **Right:** Comparison of M3GN and MGN on a special dataset. Using five different material parameters, we train five MGN models, each on a single material, and evaluate them on unseen initial conditions while assuming the material is known from the training set. As a comparison, we train a single M3GN model on the combined dataset of all five materials, where the model estimates the material properties from the context set.

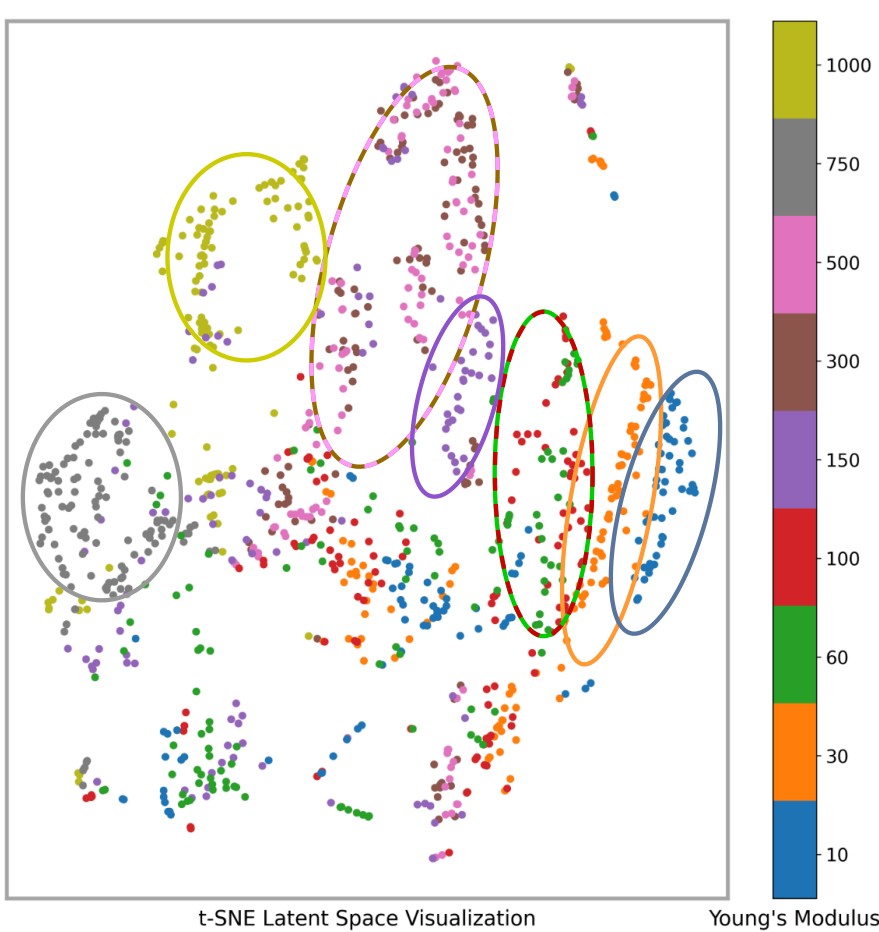

Figure 20: This figure shows a latent space visualization of the Sheet Deformation task for trajectories with 9 different Young's Modulus values, using a context size of 10. Each dot represents a 64-dimensional latent node vector projected to 2D using the t-SNE algorithm (van der Maaten & Hinton, 2008). Dots of the same color correspond to latent node descriptions for the same task, each simulated with a unique Young's Modulus. The visualization reveals distinct clustering in the latent space, with similar material properties grouped closer together, highlighting the relationship between material characteristics and the learned task representations. To improve clarity, points corresponding to nodes on the plate's edge were excluded, as their constant boundary condition resulted in unvarying latent descriptions.

## Sheet Deformation

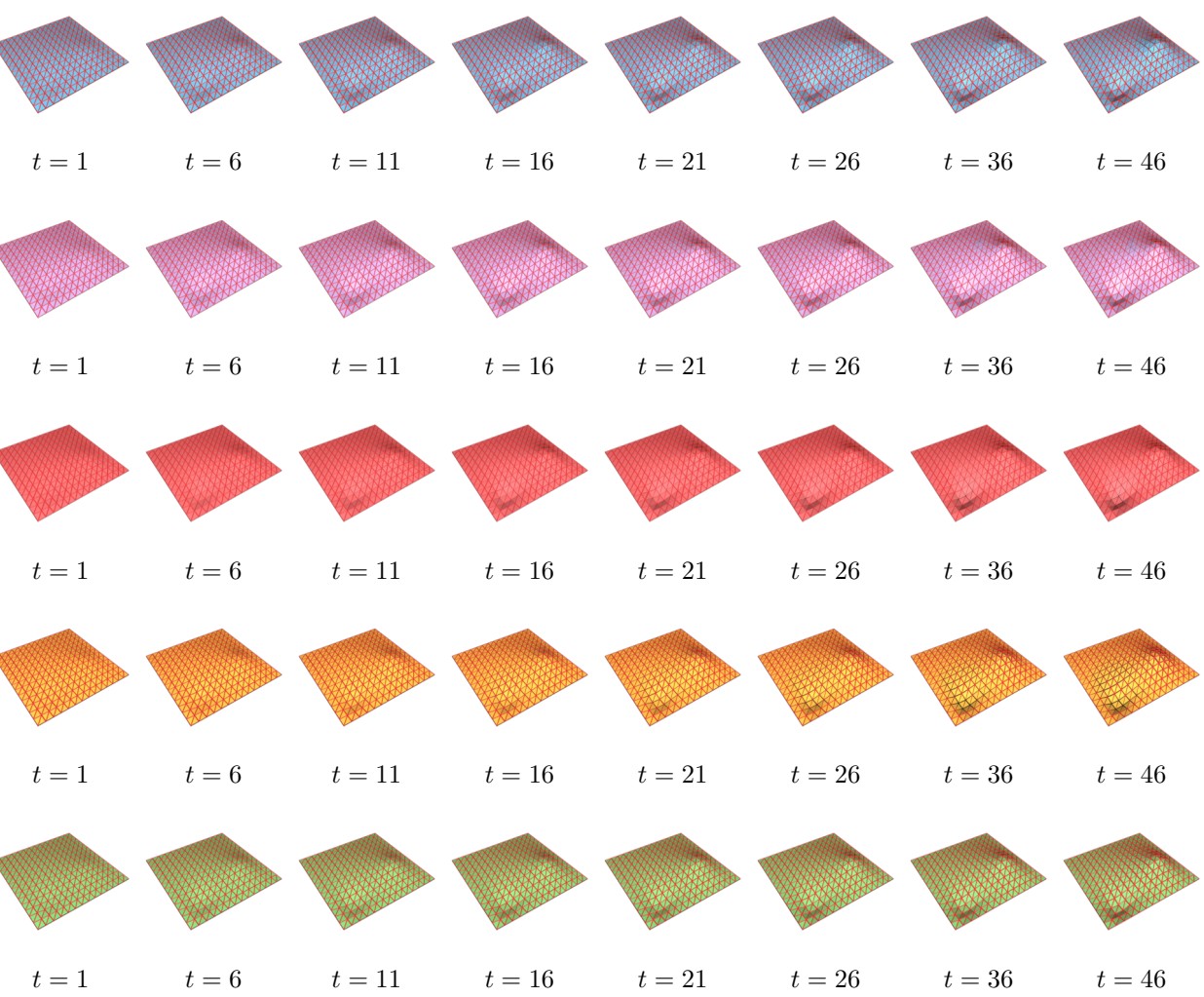

Figure 21: Simulation over time of an exemplary test trajectory from the **Sheet Deformation** task by M3GN (blue), EGNO (purple), MaNGO (red), MGN (orange), and MGN with material information (green). The **context set size** is set to 5. All visualizations show the colored **predicted mesh**, a **collider or floor**, and a **wireframe** (red) of the ground-truth simulation. M3GN can accurately predict the correct material properties, resulting in a highly accurate simulation.

# Deformable Block

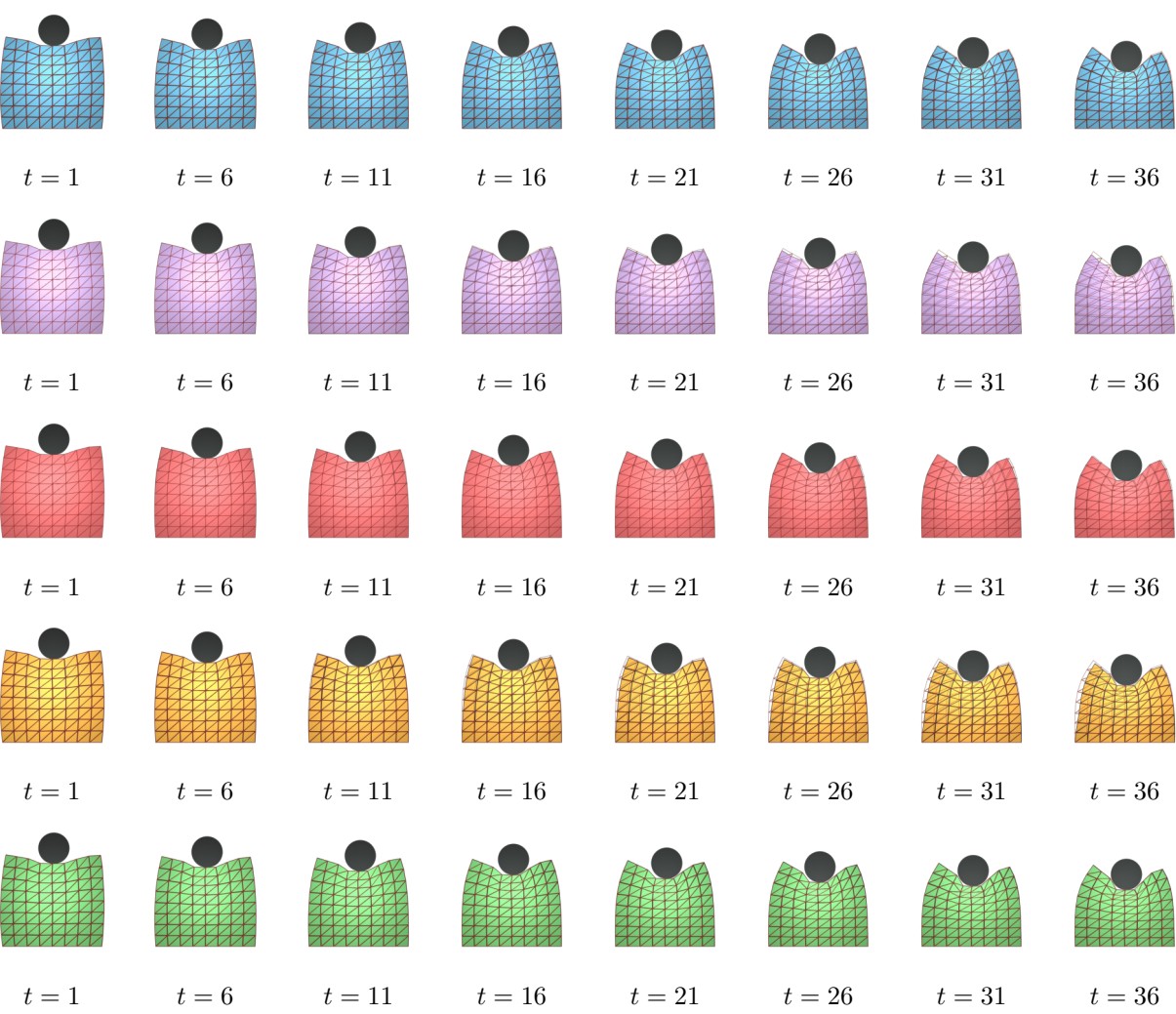

Figure 22: Simulation over time of an exemplary test trajectory from the **Deformable Block** task by M3GN (blue), EGNO (purple), MaNGO (red), MGN (orange), and MGN with material information (green). The **context set size** is set to 5. All visualizations show the colored **predicted mesh**, a **collider or floor**, and a **wireframe** (red) of the ground-truth simulation. M3GN can accurately predict the correct material properties, resulting in a highly accurate simulation.

## Tissue Manipulation

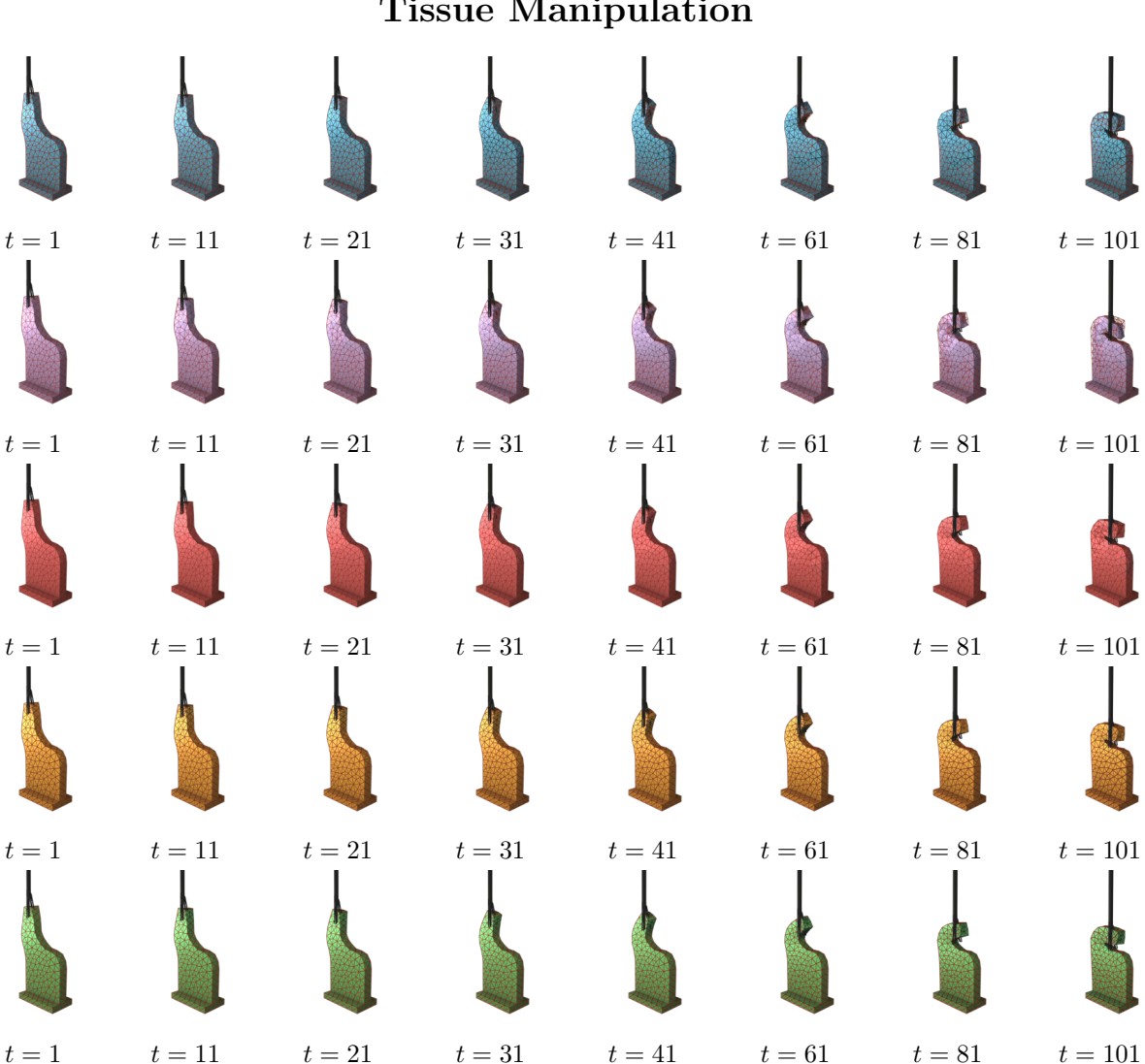

Figure 23: Simulation over time of an exemplary test trajectory from the **Tissue Manipulation** task by M3GN (blue), EGNO (purple), MaNGO (red), MGN (orange), and MGN with material information (green). The **context set size** is set to 5. All visualizations show the colored **predicted mesh**, a **collider or floor**, and a **wireframe** (red) of the ground-truth simulation. All methods can solve the task, however MGN is drifting a tiny bit to the left over time.

# Falling Teddy Bear

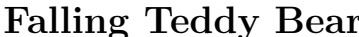

$t = 21$   $t = 41$   $t = 61$   $t = 81$   $t = 101$   $t = 121$   $t = 141$   $t = 181$

$t = 21$   $t = 41$   $t = 61$   $t = 81$   $t = 101$   $t = 121$   $t = 141$   $t = 181$

$t = 21$   $t = 41$   $t = 61$   $t = 81$   $t = 101$   $t = 121$   $t = 141$   $t = 181$

$t = 21$   $t = 41$   $t = 61$   $t = 81$   $t = 101$   $t = 121$   $t = 141$   $t = 181$

$t = 21$   $t = 41$   $t = 61$   $t = 81$   $t = 101$   $t = 121$   $t = 141$   $t = 181$

Figure 24: Simulation over time of an exemplary test trajectory from the **Falling Teddy Bear** task by M3GN (blue), EGNO (purple), MaNGO (red), MGN (orange), and MGN with material information (green). The **context set size** is set to 20. All visualizations show the colored **predicted mesh**, a **collider or floor**, and a **wireframe** (red) of the ground-truth simulation. M3GN significantly outperforms both step-based baselines.

# Mixed Objects Fall (Bunny)

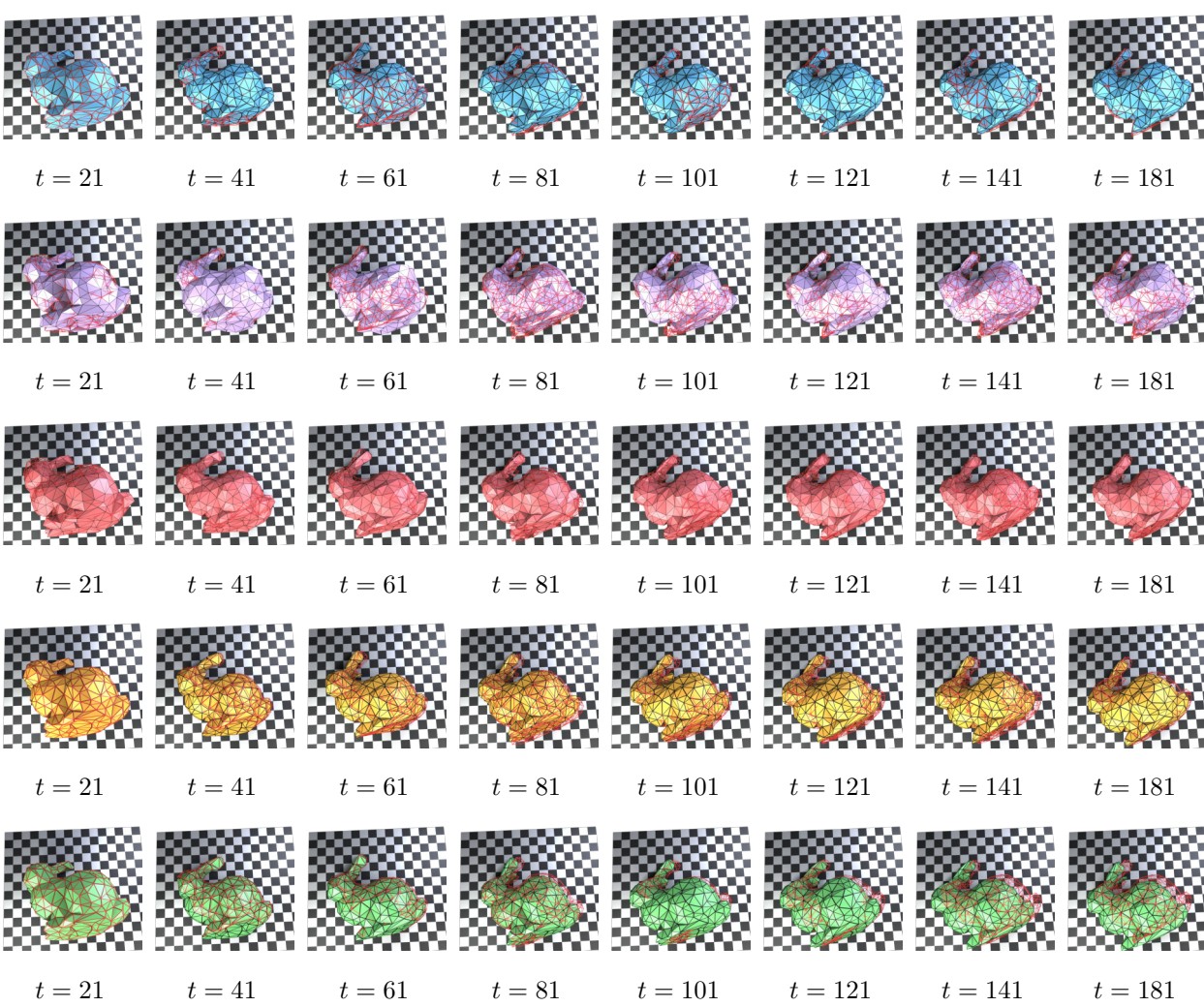

Figure 25: Simulation over time of a bunny from the **Mixed Objects Fall** task by M3GN (blue), EGNO (purple), MaNGO (red), MGN (orange), and MGN with material information (green). The **context set size** is set to 20. All visualizations show the colored **predicted mesh**, a **collider or floor**, and a **wireframe** (red) of the ground-truth simulation. M3GN significantly outperforms both step-based baselines.

# Mixed Objects Fall (Traffic Cone)

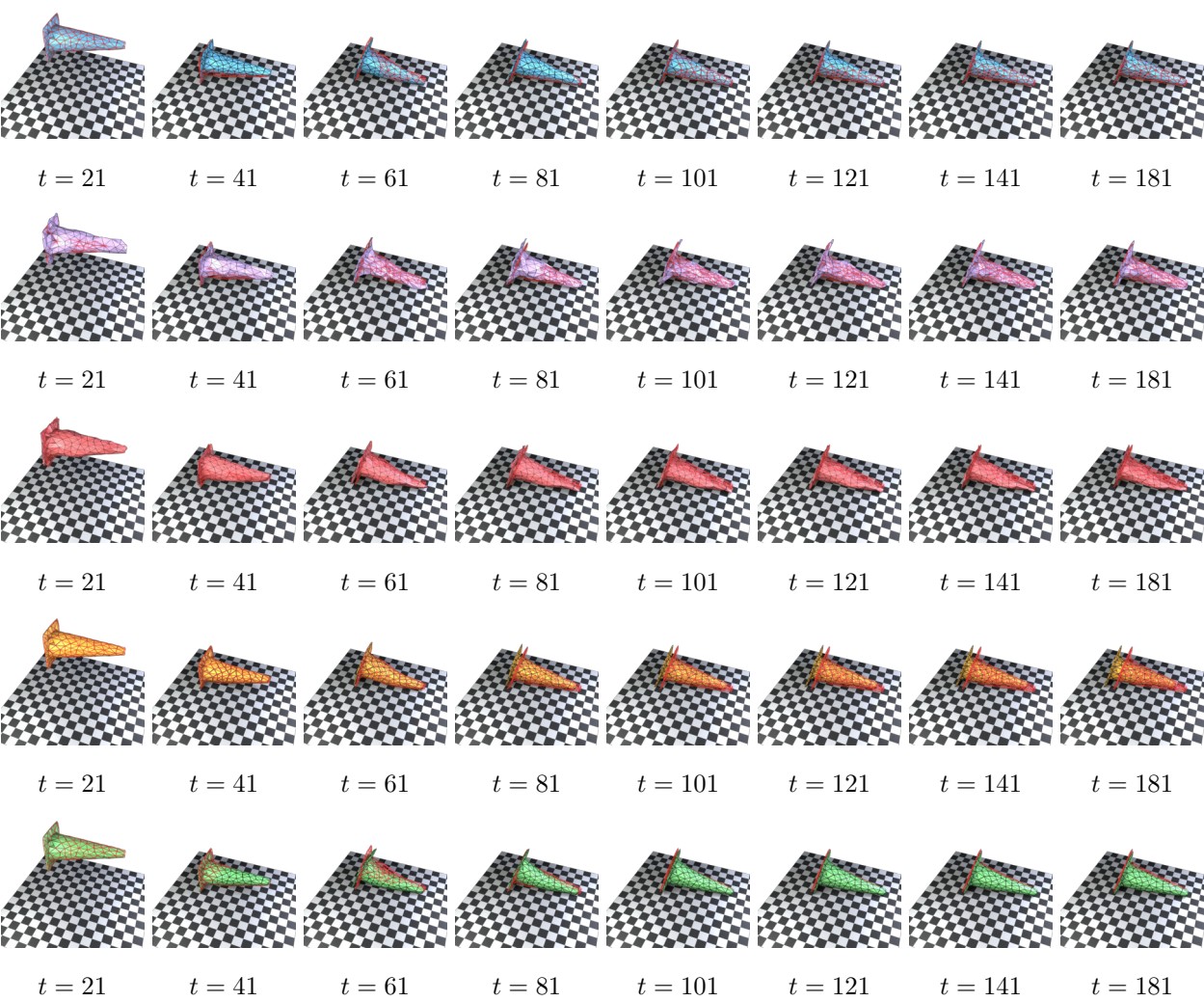

Figure 26: Simulation over time of a traffic cone from the **Mixed Objects Fall** task by M3GN (blue), EGNO (purple), MaNGO (red), MGN (orange), and MGN with material information (green). The **context set size** is set to 20. All visualizations show the colored **predicted mesh**, a **collider or floor**, and a **wireframe** (red) of the ground-truth simulation. M3GN significantly outperforms both step-based baselines. The simulation generated by MGN is severely affected by drift.

Table 3: Runtime, memory, and parameter comparison of all methods on the four benchmark tasks.

| Task | Metric | **M3GN** (Ours) | MGN | EGNO | Mango |
|------|--------|-----------------|-----|------|-------|
| Deformable Block | Context Encoding Time [s] | 0.0062 | 0.0000 | 0.0000 | 0.0000 |
| | Simulation Time [s] | 0.0132 | 0.2673 | 0.0134 | 0.0628 |
| | Total Inference Time [s] | 0.0194 | 0.2673 | 0.0134 | 0.0628 |
| | GPU Memory Usage [MB] | 68.33 | 26.94 | 121.43 | 126.10 |
| | # Parameters | 2 542 847 | 1 259 394 | 1 000 026 | 2 990 082 |
| Sheet Deformation | Context Encoding Time [s] | 0.0066 | 0.0000 | 0.0000 | 0.0000 |
| | Simulation Time [s] | 0.0132 | 0.3177 | 0.0242 | 0.1247 |
| | Total Inference Time [s] | 0.0198 | 0.3177 | 0.0242 | 0.1247 |
| | GPU Memory Usage [MB] | 170.27 | 29.12 | 210.69 | 324.03 |
| | # Parameters | 2 546 206 | 1 259 779 | 999 258 | 2 990 211 |
| Tissue Manipulation | Context Encoding Time [s] | 0.0083 | 0.0000 | 0.0000 | 0.0000 |
| | Simulation Time [s] | 0.0135 | 0.7531 | 0.0802 | 0.4705 |
| | Total Inference Time [s] | 0.0218 | 0.7531 | 0.0802 | 0.4705 |
| | GPU Memory Usage [MB] | 569.70 | 45.40 | 759.84 | 748.71 |
| | # Parameters | 2 547 614 | 1 259 907 | 1 000 794 | 2 990 723 |
| Mixed Objects Falling | Context Encoding Time [s] | 0.0354 | 0.0000 | 0.0000 | 0.0000 |
| | Simulation Time [s] | 0.0143 | 1.1961 | 0.1347 | 0.7442 |
| | Total Inference Time [s] | 0.0497 | 1.1961 | 0.1347 | 0.7442 |
| | GPU Memory Usage [MB] | 1340.79 | 42.56 | 1406.92 | 1405.34 |
| | # Parameters | 2 545 950 | 1 259 651 | 999 130 | 2 990 083 |

