# OpenReview forum: "Context-aware Learned Mesh-based Simulation via Trajectory-Level Meta-Learning"
_TMLR — Accepted by TMLR_

### Review · Reviewer_F6am · 2025-11-22

**Summary Of Contributions:**

This work proposes M3GN, a new type of Graph Network Simulator (GNS) that combines trajectory-level meta-learning and probabilistic dynamic movement primitives to solve the problem of simulating deformable objects. Using Conditional Neural Processes (CNP) to enable faster meta-learning, the proposed method resulted in magnitude times faster simulation compared to related works and showed SOTA performance on various simulation tasks. Extensive ablation studies have also been conducted.

**Strength**

- Propose to incorporate the full trajectory, rather than fixed-length steps. The proposed method thus avoids the error accumulation problem in previous step-based methods.
- Using a meta-learning framework allows the proposed method for faster adaptation to newer tasks, while outputting the full trajectory allows faster inference, resulting in a very fast simulator compared to previous works.
- Adopting meta-learning to deal with the simulation tasks is novel, especially the concept of regarding each trajectory as a new task is quite interesting
- I found this work to be well-written and organised, and is easy to follow.


**Weaknesses**
- The proposed method requires some initial observations, which is not required for previous methods.
- While extensive ablation study is done on the context size/anchor steps side, none of the ablation is done in the hyperparameter side of the proposed method.
- The authors mentioned another closely related meta-learning-based GNS method named MaNGO. However, the proposed method is not compared with MaNGO in the experiments. The claim of surpassing recent GNSs' performances is a bit weakened.

**Audience:**

Yes

**Audience Explanation:**

I think this work has potential to be quite impactful. The SOTA result from the perspective of simulator inference speed and inference accuracy and stability over the longer timesteps are quite impressive.

**Broader Impact Concerns:**

Concerns are sufficiently addressed in the Broader Impact Statement section in the manuscript.

**Claims And Evidence:**

Yes

**Claims Explanation:**

For the most part. The main contribution in this work is about proposing M3GN, a novel GNS that generate full trajectory in a single time, which can adapt to other material properties. The claim regarding inference speed is fast supported by empirical experiments. The last claimed contribution, regarding the proposed method outperforms recent GNSs on challenging deformation benchmarks, can be further strengthen if a closely related work MaNGO is also compared.

**Requested Changes:**

**Clarity/Readability**:
- C1) Very minor but, for figure captions, I would prefer also writing the colours out: for example, M3GN (blue), MGN (orange) to improve readability.
- C2) There are some inconsistent citations style across the manuscript, for example in “provides superior simulation accuracy compared to several variants of MeshGraphNet (MGN) Pfaff et al. (2021)”, “Pfaff et al.” was not parenthesised. Should be (“Pfaff et al.)” like other references. I’m sure there are more.
- C3) I might have missed but the actual basis functions used are not described anywhere.
- C4) Examples of material property can be included for clarity.
- C5) For Figure 5 (and other figures presenting the results), I’m a bit confused by the scale of y-axis. The caption notes log-scale MSE, but the plots have 1e-6/1e-4 on it. Can the authors confirm that these are indeed correct? I suspect one is written incorrectly, that is, either it’s actually MSE (i.e. the mistake is in the caption) or the mistake is in the plots.
- C6) For Figures 20-25, the meaning of the purple row (2nd row) is not described.


**Suggestions**
- S1) How is physical proximity used/determined? It’s explained in the manuscript but it’s not used/explained anywhere.
- S2) Comparison with MaNGO should be included.
- S3) Intuitively, given the usage of full trajectory of proposed M3GN, one would think that with the increasing anchor steps would naturally increase performance. This is the case for most tasks, however for mixed objects falling and falling teddy bear, increasing the anchor steps too much would actually be a bit harmful. Would be nice to have some discussion about this result.
- S4) What is the longest timestep the proposed method can be suitable for? Similar to my previous point, to my intuitive understanding, generally longer context would be more beneficial, I’m a bit confused on why one would try to input a shorter context size. Some discussion on the limitation of M3GN if possible regarding context size would be interesting.
- S5) The authors mentioned that M3GN struggles to capture rapid, high-frequency dynamics when node velocities change abruptly. It would be interesting if there are more empirical results showing this struggle.
- S6) If I understand correctly, for OOD material of the Sheet Deformation task, the scale of MSE is 10 times more than its ID counterpart, depending on the given context size. I wonder the actual meaning of this; It would be interesting to see the OOD results on other tasks as well if possible.

---

> ### Author Response · Authors · 2025-12-03
> **Rebuttal**
>
> We thank the reviewer for the positive assessment and for the thoughtful suggestions that help strengthen the paper. Below we address all points in detail and describe the corresponding updates, both planned and already integrated, in the revised version.
>
> ---
>
> Hyperparameter Ablations
>
> We agree that a large portion of our ablations focuses on context size and anchor steps. However, we would like to clarify that **a hyperparameter-related ablation is already included in the initial paper** (Appendix, Figure 12): we investigate the effect of **adding velocity inputs** for both M3GN and MGN, which directly influences model capacity, stability, and the type of information available during prediction. We have updated the text to highlight these important findings.
>
> In addition, in Figure 8, we conducted a second method-level ablation examining **different context aggregation mechanisms**, comparing our default max-based CNP aggregation with a Transformer-based aggregator. Together, these studies provide further insight into how architectural and input-related hyperparameters affect M3GN’s performance.
>
> ---
>
> S1) Use of Physical Proximity
>
> Physical proximity is used to define **edges between the collider and the deformable object**. We use a neighborhood graph based on physical proximity, choosing to connect a mesh and a collider node when their world distance is smaller than a constant threshold.
>
> Without these edges, the mesh nodes would have no mechanism to receive information about the approaching collider. This construction follows the standard approach introduced in the original MGN work.
>
> ---
>
> S2) Comparison to MaNGO [1]
>
> We agree that adding MaNGO [1] as a meta-learning baseline strengthens the paper. While MaNGO uses different data modalities, we adapted it to our setting as follows:
>
> - Only use the MaNGO decoder, as the encoder captures data over different trajectories which is not available in our setup
> - Insert the **initial context frames** into the beginning of MaNGO’s required “initial trajectory,”
> - Repeat the **anchor frame** for the remaining steps,
> - Allowing MaNGO to operate with a compact, context-like input while preserving its decoder architecture.
>
> We believe this is the fairest and most truthful adaptation of MaNGO to our problem formulation. We have run the adapted method and will include the results and a detailed discussion in the revision. MaNGO outperforms all methods on the Falling Teddy Bear task, but is worse on all other tasks compared to M3GN. Notably, MaNGO fails to extrapolate to unseen material properties in the *Sheet Deformation (OOD)* task.
>
> ---
>
> S3) Why More Anchor Steps Can Hurt (Mixed Objects / Falling Teddy Bear)
>
> The reviewer’s intuition is correct that usually a later anchor step makes the task simpler. In these two tasks, the **early part of the trajectory is trivial** (free fall), and the challenging portion begins only once the object interacts with the ground.
>
> We suspect that as the number of anchor steps increases, the easier prefix of the trajectory becomes shorter, and the metric (mean over all steps) places proportionally more weight on the harder segment. This results in a slight increase in the averaged MSE for trajectory-based methods. In contrast, step-based methods do not lose performance when using later anchor steps. Their main challenge in these tasks is controlling drift induced by the autoregressive rollout, and a shorter remaining trajectory mitigates this effect, likely offsetting the metric artifact.
>
> ---
>
> S4) Context Length vs. Prediction Horizon
>
> We clarify two distinct concepts:
>
> **Longer context** (more observed steps): In principle offers richer information for estimating material properties, and is generally beneficial.
>
> **Longer prediction horizon** (more future steps): Increases difficulty for any learned simulator, espacially for autoregressive models due to error accumultation. ProDMPs mitigate error accumulation, but extremely long horizons still introduce challenges, especially in partially chaotic settings.
>
> ---
>
> S5) High-Frequency Dynamics and Failure Cases
>
> In **Sheet Deformation** (Fig. 14), the system transitions abruptly from zero to high velocity. ProDMPs enforce smooth velocity changes and therefore temporarily oversmooth this transition, resulting in elevated error around time steps 2–5. Once velocities stabilize, the error decreases again.
>
> The issue largely disappears for context sizes ≥2, where the abrupt change is already included in the context window. We discuss this behavior in Section 4, subsection “Additional Experiments” and note that using more basis functions may reduce the effect, though we avoided excessive fine-tuning to maintain a fair comparison with the baselines.

---

> > ### Author Response · Authors · 2025-12-03
> > **Rebuttal  (Page 2)**
> >
> > ---
> >
> > S6) OOD Generalization
> >
> > For Sheet Deformation, OOD material parameters (extremely low or high Young’s modulus) yield roughly 10x larger MSE. Two factors contribute:
> >
> > - These materials lie outside the distribution seen during training.
> > - High-modulus materials produce **larger and faster deformations**, increasing the MSE.
> >
> > We agree that additional OOD experiments strengthen the paper. We plan to include a second OOD study of the deformabel block task in the camera-ready version; due to compute constraints, these results will not be ready within the rebuttal window, but we will notify the reviewer once completed.
> >
> > ---
> >
> > Clarity and Readability
> >
> > - **C1 & C2:** Updated figure captions to include explicit color references and corrected inconsistent citation formatting.
> > - **C3:** Clarified that ProDMP basis functions are obtained through **precomputed numerical integration** and do not have closed-form expressions.
> > - **C4:** We will add a figure illustrating the effect of different material properties on Sheet Deformation for the final version of the manuscript.
> > - **C5:** Confirmed that the plots show **MSE on a log-scaled y-axis**, not log-MSE. Updated caption.
> > - **C6:** Corrected the missing explanation for the purple row in Figures 20–25 (it was EGNO) and added visualizations for MaNGO for completeness.
> >
> > ---
> >
> > We thank the reviewer again for the thorough and constructive feedback. The manuscript has been improved with additional ablations, clearer explanations, expanded baselines (including MaNGO), and enhanced clarity in figures and citations. We believe these updates significantly strengthen the quality and completeness of the work.
> >
> > [1]  Mango — Adaptable Graph Network Simualtors via Meta-Learning, Dahlinger et al. (2025)

---

### Review · Reviewer_D5jb · 2025-11-22

**Summary Of Contributions:**

This paper proposes Movement-Primitive Meta-MeshGraphNet (M3GN), a context-aware, mesh-based learned simulator that leverages trajectory-level meta-learning and Probabilistic Dynamic Movement Primitives (ProDMPs) to improve the accuracy and efficiency of physical deformation simulations. Instead of performing step-wise autoregressive rollouts like conventional Graph Network Simulators (GNS), M3GN predicts full trajectories in a single forward pass conditioned on a few initial context states. This design allows the model to infer latent material properties and to avoid cumulative rollout errors.

Strengths: Innovative integration of meta-learning and movement primitives; strong empirical results with substantial runtime gains.

Weaknesses: The method’s applicability is somewhat narrow, and the model design is relatively complex to reproduce.

**Audience:**

Yes

**Audience Explanation:**

The topic is directly relevant to a portion of TMLR’s audience, particularly those working on physics-informed machine learning, graph-based modeling, and efficient simulation methods.

**Broader Impact Concerns:**

The paper includes a brief Broader Impact Statement noting potential positive impacts in reducing computational costs for physics simulations and possible misuse in developing advanced weapon models. Overall, the statement is adequate.

**Claims And Evidence:**

Yes

**Claims Explanation:**

The evidence in the paper aligns well with its main claims.
First, the quantitative results across five benchmarks (Deformable Block, Sheet Deformation, Tissue Manipulation, Falling Teddy Bear, and Mixed Objects Falling) consistently show that M3GN outperforms strong baselines such as MeshGraphNet (MGN) and the Equivariant Graph Neural Operator (EGNO) in terms of rollout accuracy and stability.
Second, the runtime analysis clearly supports the claim of improved efficiency—M3GN achieves up to 32× faster inference than MGN and 400× faster than traditional simulators due to its trajectory-level prediction and ProDMP formulation.
Finally, ablation studies (Figures 7–8) confirm that each design choice—meta-learning, ProDMPs, and CNP-based context aggregation—contributes meaningfully to performance.
Together, these results provide convincing and well-documented evidence for the paper’s claims of higher accuracy, speed, and robustness in learned mesh-based simulation.

**Requested Changes:**

- The method assumes access to future collider information at inference (e.g., inclusion of the final collider position and the note that providing the full future collider trajectory did not help). Please state explicitly what is available in realistic settings, and, if possible, add a variant where only partial/uncertain collider motion is known, with quantitative impact.
- Training assumes deterministic simulations and fixes $\sigma_o=1$ (i.e., no output variance). Add a controlled study with synthetic noise or missing context frames to quantify robustness and failure modes.
- Fig. 9 is compelling; consider adding parameter counts, peak memory, and a breakdown of time spent in context encoding vs. trajectory decoding to substantiate the “single-pass” advantage and the stated 32×/400× speedups.
- Consider adding a short discussion on how the proposed framework might generalize to non-deformation or non-mesh-based simulations, clarifying the method’s broader relevance.

---

> ### Author Response · Authors · 2025-12-03
> **Rebuttal**
>
> We thank the reviewer for the positive assessment of our contributions and for the constructive suggestions. Below we address each requested change in detail and describe the additions we have made to the revised manuscript.
>
> ---
>
> 1. Access to Future Collider Information
>
> We agree that the assumptions regarding collider information should be stated more explicitly.
>
> In realistic settings, such as robotic manipulation or planning, ****the agent typically *does* know its intended future actions (e.g., end-effector trajectories). In these domains, the collider corresponds to the robot-controlled object, and its planned future motion is available because it is usually directly commanded by the robot. The unknown quantity is the **deformable object response,** which M3GN predicts.
>
> To clarify this, we revised the manuscript to explicitly describe what collider information is assumed available at inference.
>
> ---
>
> 2. Robustness Under Noise or Missing Context
>
> While our training uses simulation data, which is deterministic, we agree that a controlled robustness study is interesting and important to assess M3GN’s application to more realistic scenarios.
>
> We added **Gaussian i.i.d. noise** to each node position in the context frames and evaluated M3GN across noise levels, both for the training and the in-context data.
>
> - Experiments are not completely finished, we revise the paper once it is finished.
> - Preliminary results indicate that M3GN remains robust across moderate noise levels, likely because the GNN encoder aggregates information across all nodes, enabling it to infer consistent trends even when individual observations are corrupted.
>
> These additions provide a clearer picture of M3GN’s failure modes and robustness characteristics.
>
> ---
>
> 3. Additional Runtime Details: Parameter Counts, Memory, and Timing Breakdown
>
> We have expanded the runtime analysis as suggested.
>
> Table 3 in the Appendix now includes:
>
> - **Parameter counts** for M3GN, MGN, ProDMP variants, and MaNGO
> - **Peak GPU memory usage** during inference
> - A breakdown of **context encoding time** vs. **trajectory decoding time**
> - Updated timing comparisons including **MaNGO**
>
> These results further substantiate the claimed 32x speedups and highlight the efficiency gains of single-pass trajectory prediction.
>
> ---
>
> 4. Discussion on Generalization Beyond Mesh-Based Deformation
>
> We added a short discussion on how M3GN could extend to other domains:
>
> - **Non-deformation tasks (e.g., fluids or particle-based interactions):**
> The meta-learning + trajectory-primitive structure applies directly, since these tasks also involve predicting long-horizon physical responses conditioned on initial states. The removal of contact dynamics make the ProDMP approach even more appealing.
> - **Non-mesh-based simulations:**
> M3GN expects graph-structured inputs, but this requirement can often be met via particleization or mesh extraction pipelines. Extending the approach to non-graph inputs would require additional architectural modifications but remains conceptually feasible. A point cloud input could be converted to mesh data [1], while image input data might require further processing steps.
>
> This clarifies the broader relevance of the framework while being realistic about its current limitations.
>
> ---
>
> We thank the reviewer again for the thoughtful feedback. The new robustness study, expanded runtime analysis, clarifications on collider assumptions, and discussion of broader applicability meaningfully strengthen the paper.
>
> [1] C. Lv, W. Lin and B. Zhao, "Voxel Structure-Based Mesh Reconstruction From a 3D Point Cloud," in *IEEE Transactions on Multimedia*, vol. 24, pp. 1815-1829, 2022

---

> > ### Author Response · Authors · 2025-12-16
> >
> > We would like to update the reviewer that the robustness ablation with respect to noisy context data has now been fully completed and is included in the current revision of the manuscript. The finalized results are presented in the Appendix (Figure 19) and analyze M3GN’s performance across different noise levels in both the training and in-context data. These results confirm the trends discussed previously and provide a complete and systematic characterization of M3GN’s robustness to noisy context observations.

---

> > > ### Comment · Reviewer_D5jb · 2025-12-18
> > >
> > > I hereby confirm that I've read the updates provided by the authors. I have no more concerns regarding this submission. Thanks for the good work of the authors.

---

### Review · Reviewer_ifmq · 2025-11-22

**Summary Of Contributions:**

The paper describes a novel problem formulation of deformable object simulation as meta-learning and provides a model architecture and training methodology for that problem. They show that their method (M3GN) is able to leverage the additional context effectively to produce more accurate simulations than a baseline method that is directly given material properties about the object to be simulated.

Specifically, they make the following contributions:

* Propose a Graph Network Simulation algorithm that 1) uses the initial context to learn latent descriptors; and 2) outputs the weights of Probabilistic Dynamic Movement Primitives (ProDMPs) for each node.
* Evaluate this method against several baselines and demonstrate effective performance.

One important comment is that the authors don't explicitly claim that the problem definition is a contribution; however, I think it needs to be (or, at least, the introduction of a novel problem setting needs to be more heavily emphasized). Otherwise, it is not clear that their comparisons are to baselines that are not designed to generalize across different deformable object parameters.

## Strengths
* The problem formulation seems realistic and allows for simulators to give improved performance without needing to solve the real-to-sim-to-real problem.
* The use of ProDMPs to model trajectories is interesting/novel and seems to potentially improve the quality of simulation over long time horizons. It also seems well-suited to efficient optimization.

## Weaknesses
* The paper isn't very clear (to my reading) that the baselines are being applied to a task they aren't really designed for. This is a new problem and the authors are attempting to adapt the baselines to a new problem setting. This is a reasonable choice but needs to be made much clearer to readers as outperforming the baselines isn't as impressive in this context. It would be good to see the authors compare M3GN's performance with that of MGN (or other baselines) when trained on a single deformable object. This would allow evaluation of how well the meta-learning performs in comparison to a skyline method that stacks the deck in favor of MGN.

* The paper doesn't explicitly compare against Dahlinger et al. 2025, which seems like the closest method from related work and the primary other method to apply meta-learning to a similar problem. The discussion in related work simply states that their approach relies on different information (a distinct trial vs a prefix of a trajectory) and is less memory efficient (but the actual comparison for memory costs is not made in the paper). At a minimum, it would be good to more clearly describe the experimental rationale and the pro/cons of the different baselines to make it clear how each baseline adapts to the meta-learning problem.

* The evaluation confounds the benefits that come from two novel changes to the MGN approach: the learned latent parameters based on the context and the ProDMP parameterization of the trajectory. It would be nice to see comparisons that separate out the benefit from each of these additions to the baseline. For example, does MGN with ProDMP output parameterization see substantial performance on its own? If this ablation does not make sense, then the authors should clearly explain why this experiment isn't appropriate.

* The paper is technically dense, and this makes it hard to identify the key points. (I personally had to spend a lot of time going back and forth and it took several passes through the paper before I felt like I understood the main ideas.) For example, the introduction is almost 3 pages long and first discusses GNS broadly, then discusses error accumulation over time, then introduces an example task that is really about the availability of initial context, then jointly introduces meta-learning and trajectory-level predictions, then describes ProDMPs, and then describes the method. Better clarity in the writing would help substantially. Consider, e.g., a structure that explains 1) why GNS is useful; 2) that it needs to train from scratch for a new material; 3) why auto-regressive prediction is problematic, how other methods have tried to fix the problem and the associated drawbacks for memory... etc; 4) how M3GN fixes both of these problems by meta-learning and outputting ProDMP parameters.

* The paper claims that M3GN is faster than MGN, but doesn't actually present evidence for the speed of the different methods. Similarly, the authors claim that ProDMP as a trajectory parameterization provides memory efficiency over alternative trajectory transformations but don't substantiate this with evidence.

* The related work section currently mixes general background discussion of methods with comparison to/differentiation from related SOTA  methods. This makes it harder to understand what is novel and assess the work.

## Overall Assessment

I think this paper has a lot of potential and I would like to see it accepted. However, as it stands there

**Additional Comments:**

I found the description of ProDMP in the appendix quite helpful --- it might be good to reference that more explicitly in the main body of the text so that readers unfamiliar with ProDMP are aware of the resource.

**Audience:**

Yes

**Audience Explanation:**

The problem considered is important and interesting and the results are promising. Overall, I expect that this paper would be interesting a range of TMLR readers who are interested in learned differentiable simulation.

**Claims And Evidence:**

No

**Claims Explanation:**

The claims about the overall performance of the method are well substantiated with experiments in comparison to MGN. However, there are some missing pieces of evidence:

* comparison with a relevant meta-learning approach with respect to both prediction performance and cost (time/memory) are not presented and improvement is claimed without substantiating evidence
* the authors claim that M3GN is 32x faster than MGN but don't directly support this

The second issue seems like a relatively minor fix. The first one likely requires additional experiments to be run (or a clear description of why the comparison is inappropriate). I understand that MaNGO relies on different context, but I think it is reasonable for the authors to modify MaNGO to fit their setting (or to run experiments that match the configuration for MaNGO).

**Requested Changes:**

* Improve clarity of the presentation that makes it clear which problems are being solved and separates the changes to the problem setup from the description of the solution proposed.
* Experiments that more clearly isolate the distinction between the benefits of the meta-learning approach presented and the contribution from the trajectory representation of ProDMPs as the output of the network. Ideally, the authors would produce ablations that independently assess the benefit of these changes in comparison to MGN.
* Comparisons with MaNGO or a more explicit discussion of why the comparison is inappropriate
* Experiments that explicitly measure the performance gains claimed with respect to speed and memory requirements of the different methods
* Experiments that compare the performance of M3GN with MGNs trained on a fixed set of mesh dynamics would be nice to gauge the relative performance of M3GN in comparison to a more costly approach of re-training an MGN for each object. This would substantially help to calibrate the results and improvements.
* Changes to the related work that more clearly distinguish the discussion of background methods that this work builds on from comparing and contrasting this work with alternative problems or methods. Consider, e.g., splitting this section into a background section and a more direct related works section.

---

> ### Author Response · Authors · 2025-12-03
> **Rebuttal**
>
> We thank the reviewer for the detailed, insightful feedback. We address each point below and describe the concrete changes and additional experiments we have already made or will include in the revision.
>
> ---
>
> 1. Clarifying the Novel Task Setting and Baseline Adaptation
>
> We agree that the novelty of the task setting needs to be made clearer. In the revision, we explicitly highlight that we introduce a new **trajectory-based meta-learning** setting where models must generalize across varying material parameters using only initial trajectory context.
>
> Regarding the claim that baselines were “not designed” for this setting, we clarify that **MGN with Material Information** actually has privileged access to exactly the information needed to solve the task. Additionally, we adapted MGN (already in the submitted initial version of the paper) by providing **the last velocity** as additional input, ensuring it has at least the direct context of the anchor step. Larger velocity windows were also tested but led to overfitting. This adaptation substantially improves baseline performance, e.g., for **Falling Teddy Bear** (Appendix Fig. 12). We argue that this results in a fair comparison on our experimental setup.
>
> ---
>
> 2. Comparing M3GN to MGN Trained on a Single Material (Skyline Baseline)
>
> We appreciate the suggestion to compare M3GN to per-material MGNs. We propose the following experiment, which we believe captures the reviewer’s intended  experimental setup.
>
> Select **5 distinct material parameters, unlike the current continuous range of parameters**.
>
> 1. For each material: generate 1/5 of the original training data, so **140 train** and **30 test** trajectories.
> 2. Train M3GN on the combined dataset (on 700 trials, same size as our current datasets).
> 3. Train **five separate MGNs** (on 140 trials per material).
> 4. Evaluate each test trajectory using the corresponding MGN.
>
> With this, each MGN can focus on its learned material, and has to interpolate or generalize to new parameters. We are happy to run this experiment, or change the setup as requested by the reviewer. Due to dataset generation and training time, results will not be ready by the end of the rebuttal phase, but they will be included in a future revision/camera-ready version.
>
> ---
>
> 3. Comparison to MaNGO [1]
>
> We agree that adding MaNGO [1] as a meta-learning baseline strengthens the paper. While MaNGO uses different data modalities, we adapted it to our setting as follows:
>
> - Only use the MaNGO decoder, as the encoder captures data over different trajectories which is not available in our setup
> - Insert the **initial context frames** into the beginning of MaNGO’s required “initial trajectory,”
> - Repeat the **anchor frame** for the remaining steps,
> - Allowing MaNGO to operate with a compact, context-like input while preserving its decoder architecture.
>
> We believe this is the fairest and most truthful adaptation of MaNGO to our problem formulation. We have run the adapted method and will include the results and a detailed discussion in the revision. MaNGO outperforms all methods on the Falling Teddy Bear task, but is worse on all other tasks compared to M3GN. Notably, MaNGO fails to extrapolate to unseen material properties in the *Sheet Deformation (OOD)* task.
>
> ---
>
> 4. Ablations: Separating Meta-Learning and ProDMP Effects
>
> We agree that isolating the individual contributions of both meta-learning and the ProDMPs is essential. These experiments are already included in Fig. 7:
>
> - **MGN(MP)**: ProDMP output without meta-learning.
> - **M3GN(step)**: Meta-learning without ProDMPs.
>
> Evaluations on SD and DB show that **each component alone provides limited benefit**, while their combination yields the largest gains. We are happy to run additional parameter studies if further specified by the reviewer
>
> ---
>
> 5. Paper Structure and Exposition
>
> We appreciate the suggestion and streamlined the introduction in the revision. The reorganization significantly improves readability.
>
> ---
>
> 6. Speed and Memory Claims
>
> The initial submission already included a speed comparison (Fig. 9). We additionally extended our evaluation to include **memory consumption** (including MaNGO), and the full results will be added to Appendix Table 3.
>
> Summary:
>
> - **M3GN is substantially faster than MGN**, especially on long-horizon rollouts (due to M3GN’s constant time inference compared to the O(T) time inference for the auto-regressive rollout with MGN) .
> - **M3GN is more memory-efficient than MaNGO**, due to the compact ProDMP representation.
>
> These results directly substantiate the claims made in the submission.
>
> ---
>
> We thank the reviewer again for the thoughtful feedback. We believe the clarifications, new experiments, and structural improvements substantially strengthen the paper, and we are committed to implementing the skyline baseline pending confirmation from the reviewer.
>
> [1]  Mango — Adaptable Graph Network Simualtors via Meta-Learning, Dahlinger et al. (2025)

---

> > ### Comment · Reviewer_ifmq · 2025-12-07
> > **Thank you for the thoughtful response, my concerns are suitably addressed**
> >
> > I thank the authors for their engagement and response.
> >
> > My only quibble is that, while I appreciate the modifications to include the previous velocities and material information in MGN does adapt it somewhat, I still think it's important to clarify the distinction to the readers. I appreciate the offer to compare with MGN that stacks the deck in favor of the prior method (by training an individual MGN per material). Under the assumption that this will be included in the camera-ready, I am happy to recommend accepting the paper.
> >
> > I apologize that I missed the ablation study and timing information in my initial review of the work. I recommend updating the paper to more clearly highlight those to the reader.

---

> > > ### Author Response · Authors · 2025-12-16
> > >
> > > We thank the reviewer for the careful reading and for the positive assessment of our revisions.
> > >
> > > We have added the requested comparison to MGN models trained separately for each material setting in the revised version of the paper. This comparison is now included in the Appendix, Figure 19 and will be part of the camera-ready version. The results show that M3GN achieves performance comparable to these per-material MGN models, highlighting its ability to generalize to expert, material-specific models by inferring material properties directly from the provided context data.
> > >
> > > We also appreciate the reviewer’s acknowledgment regarding the ablation study and timing information. In the revised manuscript, we have made these results more clearly visible and explicitly highlighted them to improve accessibility for the reader.

---

### Author Response · Authors · 2025-11-28
**Update on Ongoing Revisions and Upcoming Response**

Dear Reviewers,

Thank you very much for your valuable feedback and for the time you invested in reviewing our submission. We are currently working through all requested revisions and additional experiments.

Specifically, we would like to inform you that:

- We are performing the individual experiments requested in each review and will **provide detailed, point-by-point answers by the beginning of next week.**

- We have completed the integration of the **MaNGO baseline** (Dahlinger et al., 2025) and have run the corresponding experiments. The updated results will be included in our next revision (also at the beginning of next week).

- In the revision, we will also add additional information regarding **memory consumption and inference timings.**

- We have conducted a stability analysis under varying levels of noise in the context data.

We appreciate your constructive comments and look forward to submitting an improved version of the manuscript shortly.

---

> ### Author Response · Authors · 2025-12-03
> **Revision**
>
> We have updated the paper and uploaded a new revision. Changes relative to the previous version are highlighted in blue. The main additions are the comparison to MaNGO and the extended analysis of memory consumption and inference timings. We have also provided detailed responses to all reviewer comments.

---

> > ### Author Response · Authors · 2025-12-16
> > **Revision #2**
> >
> > We further improved the paper by addressing the reviewers’ suggestions and adding three additional components:
> >
> > - We added MaNGO visualizations to the Appendix (Figures 21–26) to provide qualitative insights into model behavior.
> > - We included the requested stability ablation of M3GN under noisy context data (Figure 19, left, in the Appendix).
> > - We added a comparison between M3GN and five MGN models, each trained and evaluated on a single material setting, as requested (Figure 19, right, in the Appendix).
> >
> > As before, all changes made in response to the reviewers’ comments are highlighted in blue in the rebuttal.

---

### Decision · Action_Editor_waVp · 2026-01-02

**Recommendation:** Accept as is

**Audience:**

Yes

**Audience Explanation:**

Yes, the paper is valuable for the TMLR audience interested in the research areas of physics-informed machine learning and differentiable simulation.

**Claims And Evidence:**

Yes

**Claims Explanation:**

The paper rigorously benchmarks the method and has conducted an initial study of ablation studies. Throughout the rebuttal period, a revision has further added analyses wrt run-time and memory, as well as further baseline comparisons. The initially criticized lack of support regarding efficiency-improvement statements by reviewers could thus be resolved. The revised version accurately and convincingly supports its claims. Various revisions regarding the readability and clarity of made statements have also been made in the revision, which was another main caveat initially raised by reviewers and subsequently resolved.

---

> ### Author Response · Authors · 2026-01-08
>
> We would like to thank the reviewers for their careful reading of the manuscript and for the detailed, constructive feedback provided during the review and rebuttal phases. Their suggestions significantly helped us strengthen both the presentation and the empirical evaluation. We also sincerely thank the Action Editor for their time, effort, and thoughtful management of the review process.